# Statistical approaches for service delivery differentials as assessed through a composite indicator: Application to Ugandan local governments

Hillary Muhanguzi[1]*, Francesca Bassi[2], Yeko Mwanga[3], James Wokadala[3]

**1** Department of Statistical Methods and Actuarial Sciences, Makerere University, Uganda, **2** Department of Statistical Sciences, University of Padova, Italy, **3** School of Statistics and Planning, Makerere University, Uganda

\* muhabs7@gmail.com

## Abstract

This study assesses differentials in service delivery among Ugandan local governments through a composite indicator that consolidates essential performance data from the education, health, and water sectors. The composite indicator scores as an outcome variable and it is modeled against probable determinants using beta regression, generalized additive models (GAMs) and random forest regression approaches. Beta regression, is a parametric approach, which includes varying precision parameters, is effective for modelling bounded data such as composite indicators, whereas random forest regression, as a non-parametric approach, emphasizes the relative importance of predictors. On the other hand, GAMs are semi-parametric, and bring to the fore non-linear covariate effects employing splines. Employing the minimax transformation, equal weighting and multiplicative aggregation, the composite indicator of service delivery scores varied from 0.25 to below 0.60 (on a scale of 0–1), with a substantial number of local governments scoring below 0.5. The findings reveal budgetary constraints, fragmentation at sub-county level, and geographical challenges in terms of distance from the capital city as significant obstacles to service delivery at local government level. The predictive accuracy of the three approaches as determined through the mean square error (RMSE) were found to be comparable (RMSE ≈ 0.05; MAE ≈ 0.042), suggesting that these approaches, grounded in contextualized theoretical frameworks, are effective in assessing service delivery outcomes and may therefore be employed in similar studies after careful consideration of the analytical objective. The study recommends broadening the range of service dimensions and predictors to develop a more relatable composite indicator. Given the group structures present in the predictors, employing grouped regression within the framework of beta regression modeling is advisable to provide a more robust and efficient modeling strategy.

**Data availability statement:** The data are sourced from different publications and collated by the corresponding author in two files. The two sets of data have been attached as supporting information for sharing with the journal.

**Funding:** The author(s) received no specific funding for this work.

**Competing interests:** The authors have declared that no competing interests exist.

## Introduction

Sufficient provision of services is essential to the ambitions of any developing nation, as demonstrated by national pledges in various global and domestic development frameworks. This is crucial due to its impact on enhancing citizens' quality of life, alleviating civil unrest, decreasing living costs, increasing revenues, and as posited by the Organization for Economic Cooperation and Development (OECD) [1], the quality of services is a measure of a nation's governance. The income implications are exemplified in Uganda, where the World Bank estimates an annual loss of USD 800 million (equivalent to 24,000 daily man-hours) in productivity attributable to inadequate service delivery manifested as road traffic congestion [2]. The commencement, packaging, solicitation, delivery, and fulfillment of service requirements include several stakeholders and interconnected iterative processes, making this phenomenon multi-dimensional. Due to the complexity related to measuring aspects of societal well-being [3], service delivery is captured and assessed through an array of distinct indicators that have no universal standard unit of measurement. In fact, a comprehensive, standardized set of indicators for assessing service quality is lacking, as current indicators are often fragmented and concentrate either on final outcomes or inputs, rather than on the foundational systems that produce the outcomes or utilize the inputs [4]. Moreover, there is no consensus on a standardized set of metrics to assess the limitations related to service delivery and the conduct of front-line providers, both of which directly influence the quality of services accessible to residents. Service delivery can be described as any interaction with public administration in which individuals, or firms seek or supply information, manage their affairs, or execute their obligations [1,5]. Historically, state governments predominantly deliver services including justice, healthcare, education, transportation, security, and regulatory administrative services that require legislation, permits, and enforcement. Recently, some services have been rendered by non-state actors such as private-for-profit hospitals, schools and private security. Moreover, through privatization some services such as banking, and energy are provided on behalf of the state, albeit within its policy umbrella. Regardless of the provider, for citizens to utilize these services, they must be provided in an effective, informed, accessible, predictable, reliable, and customer-oriented manner. The Ugandan Government has been focused on improving service delivery by implementing a decentralization policy [6] designed to transfer substantial administrative, political, and fiscal authority from the central government to local governments (LGs), thereby amplifying the influence of local citizens and optimizing local service delivery [7]. Consequently, there are more than 150 local governments, including districts, towns, and cities, which serve as hubs for planning and service delivery to the populace [8]. Within this framework of resource transfers, the establishment of local governments and assessments, Uganda has encountered successes and challenges in service delivery. These various issues are evidenced by a multitude of disparate service delivery metrics, indicators, and methodologies, including national service delivery surveys [9], performance expenditure tracking surveys [10], government-citizen interaction forums (*barazas*), local government performance assessments [11], evaluations of the National Development Plans (NDPs)

and non-state actors [12] program evaluation studies. While they have demonstrated efficacy in identifying bottlenecks, inefficiencies, and other issues in service delivery, they continue to produce an array of disparate indicators and statistics that hinder coherence in not only service delivery measurements but also policy discussions. Bearing in mind the multi-dimensional nature of service delivery, we constructed a composite indicator of service delivery in Uganda focusing on the three sectors of education, health and water. This attempted to consolidate the existing diverse elementary indicators into a singular, user-friendly metric that would provide significant insights into the operation of various service delivery centers, particularly the district local governments. Utilizing a comprehensive composite indicator, we analyzed it in relation to a set of potential determinants to identify associations with service delivery outcomes. Assigning service delivery performance to local governments in Uganda is justified as Article 191 of the Constitution and Section 80 of the Local Government Act (CAP 243), local governments are mandated to prepare their own development plans and budgets, mobilize revenues locally to facilitate funding for recurrent and development expenditure for service delivery [1,13]. District local governments within the devolved service sectors are responsible for determining both short- and long-term inputs, as they enact legislation, plan, provide, and monitor services. Despite financial limitations, local governments hold significant authority to impact service delivery outcomes within their jurisdictions; therefore, it is reasonable to empirically evaluate their performance.

Authors such as Satria et al. (2025) have demonstrated that a range of local government performance measurements exist owing to its multi-dimensionality and complexity, thus proposed for a contextual-based measure [14]. The Municipality Sustainability Index (MSI) developed by Caldas et al. (2020), which includes 25 indicators across five dimensions, exist; however, this MSI has not been analyzed against potential covariates [14]. Furthermore, the majority of local government performance studies have mostly focused on economic and financial performance aspects [15], frequently neglecting emerging concerns such as sustainability and gender diversity. While these indicators highlight the performance objectives of a specific local government, comparative analyses across statistical units that explore related aspects may be overlooked, particularly in developing nations. In light of the novel composite indicator developed for the Ugandan context, we associate service delivery composite indicator (CI) scores with specific predictors, primarily to ascertain the most effective statistical approach and to enhance the existing literature by illustrating that comprehensive regression analysis reveals facilitators or impediments to local government service delivery. Statistically, the complexity of service delivery and probable predictors would require the application of diverse approaches which are compared on accuracy and predictive power. Ultimately, we identified a more robust approach and employed it to elucidate the disparities in service delivery at the district local government level. Nonetheless, we acknowledge that there is no "correct" model, as any model is a simplification of reality [16]. Nevertheless, a robust model is characterized by superior predictive accuracy, which also includes the precision of forecasts derived from previously un-utilized test data.

In this paper, the introduction chronicles the building of the service delivery composite indicator, giving an overview of statistical approaches that may apply to this study. Methods, including the process of building the composite indicator are described in the second section. In the third section, we present the results while the fourth section encompasses the discussion. The paper concludes in the fifth section.

## Notation and the three statistical approaches

The statistical approach in a regression setting seeks to identify and employ the most suitable function for mapping inputs to the service delivery outcomes [17]. Before modelling, it is essential to acknowledge the various data treatments derived from univariate and bivariate studies and their implications, as early data exploration is frequently necessary to determine the best appropriate model [18]. The quantitative outcome/the composite indicator score is denoted as $Y$ and $p$ different inputs/predictors. For some input values $X = (X_1, X_2, X_3, \ldots, X_p)$, the function as in equation (1) is modeled on the assumption that there is a relationship between $Y$ and $X$.

$$Y = f(X) + \varepsilon \tag{1}$$

where $f$ is some fixed but unspecified function of $X$ and $\varepsilon$ is an *i.i.d* random variable that describes observation noise and perhaps un-modeled processes. Primarily, statistical modelling is to find a function $\hat{f}$ (either parametrically or otherwise) based on the observed data, that is close to the unknown function $f$ that generated the data and demonstrates effective generalization [17]. Parametric statistical modelling involves two steps: making of assumptions about the functional form of $f$, which facilitates model selection, and using training data to fit or train the model [19]. In contrast, non-parametric approaches do not impose explicit assumptions regarding the functional form of $f$, instead they seek an estimate of $\hat{f}$ in a manner that closely approximates the data points while maintaining smoothness. The advantage is that by avoiding the assumption of a particular functional form for $f$, they have the potential to accurately fit a broader spectrum of feasible shapes for $f$. However, this can be detrimental as it typically results in over-fitting and produces low accuracy on data not utilized for model training.

In our view, a significant challenging phase in a study is selecting an appropriate model or approach that fits a set of predictors to the dependent variable. This is because, no single method surpasses all others over all potential data sets [20,21]. As such, selecting the appropriate family of models is essential, necessitating multiple options to address various research inquiries [22]. Moreover, statistical approaches differ from decision analysis since statistical analyses may end with an indication of the probabilities and related uncertainties while the latter culminates in the selection of a single decision [23]. Considering this position, we provide a concise overview focusing on the prerequisites, possibilities and limitations of a selection of potential models applicable to an investigation of this kind. We commence with the linear regression model, according to Hastie et al. (2019) the regression function $E(Y|X)$ is linear in the inputs $X= (X_1, X_2, X_3,…,X_p)$ [20]. Furthermore, linear regression is as well linear in the parameters [17,18]. It predicts $Y$ as a weighted aggregate of the inputs, typically employing the ordinary least squares approach for estimation. Besides the linearity assumption, the linear model is premised on other assumptions, such that $\varepsilon \sim N(0, \delta^2)$ and the absence of multi-collinearity among the predictors. The strength of the linear model lies not only in its user-friendly simplicity [24] and the clarity it provides in interpreting the link between predictors and outcomes, but it also serves as a robust foundation for extensions or generalizations to other models. The disadvantage with the linear regression model is that it works well for regression but not for classification. In addition, accuracy and simplicity (interpretability) are at odds, as the accuracy of a linear model typically falls short compared to that of the less interpretable neural networks [21]. Again, if the assumptions are breached, the confidence intervals of the coefficients become unreliable. To achieve sparsity, model regularization techniques like lasso and ridge regressions extend multiple linear regressions by shrinking coefficients, resulting in a final model where the outcome is associated with just a limited subset of predictors.

This study aligns with the assertion that statistical modeling necessitates a choice between parametric and non-parametric approaches [25]. Accordingly, we select models categorized into three types: parametric (beta regression), semi-parametric (generalized additive model), and machine learning (random forest). The parametric approach employs beta regression, which is effective for modeling doubly bounded response variables while ensuring that fitted values ($\hat{y}$) remain within specified intervals. Parametric modeling, while simpler and more interpretable, has faced criticism when its strict assumptions are violated. For instance, if the true distribution is more complex than the assumed parametric models, this can result in potential under-fitting. In contrast, non-parametric modeling enables the data to be analyzed without strict assumptions, making it more suitable in cases where assumptions are violated and when outliers are present. A semi-parametric model is intermediate between parametric and nonparametric models and contains finite-dimensional and infinite-dimensional components. This study applies the generalized additive model- a semi-parametric model- in elucidating non-linear covariate effects. Recent advancements in artificial intelligence and machine learning have led to the application of various regression algorithms, offering alternative methods for achieving more objective and accurate regression modeling. Random Forest (RF) is a machine learning algorithm that mitigates over-fitting and offers a method

for assessing feature importance in the prediction process. Within the domain of machine learning algorithms, we evaluated the application of RF due to its effectiveness in modeling both classification and regression tasks, its ability to mitigate over-fitting, and its proficiency in assessing important features that can be readily communicated to stakeholders. This study demonstrates that employing multiple modeling approaches enhances robustness and mitigates the risk of findings being artifacts of a single method's assumptions, particularly when these approaches yield similar conclusions. Furthermore, due to the strengths and weaknesses inherent in each approach, a comparative analysis aids in balancing accuracy and generalizability, thereby establishing a foundation for selecting the most suitable approach for the analytical objective.

## Beta regression

Since its introduction by Ferrari and Cribari-Neto (2004), beta regression models have become prominent in modelling doubly bounded response variables inside the interval (0,1) or [0,1], that are related to the predictors through a regression structure [26]. Like our developed composite indicator, other variables in that characterization of (0,1) encompass proportions, percentages, and rates. Beta regression has previously been used to model an evenness index and a straightness index- which are composite indicators [27]. Before its development, continuous doubly bound data were exclusively modeled using transformations and a normal error distribution [28]. Linear regression, however, is unsuitable for this data: i) it may generate fitted values for the response variable that surpasses its lower and upper limits; ii) as proportions or composite indicators, they are often asymmetrical, rendering inferences based on the normality assumption potentially misleading. Furthermore, due to the data modifications especially due to angular transformation, the model parameters are not readily interpretable in relation to the original response and, angular transformation performs poorly and should be avoided in cases where the denominators or sample sizes for the variables are different [29].

Beta regression accounts for heteroscedasticity by positing that the dependent variable follows a beta-distribution characterized by the mean and the dispersion parameters. The mean is associated with a set of covariates via a linear predictor with unspecified coefficients and a link function whilst the dispersion parameter may either be fixed or vary, contingent upon the set of covariates through a link function as well. Estimation in a beta regression model is conducted via maximum likelihood, with model parameters interpretable in terms of the mean of the response and, when the logit link is used, of an odds ratio [26]. Although there are no significant differences in model diagnostics compared to the previously utilized angular transformation models, beta regression yields more reliable parameter estimates, particularly when effect sizes are regarded as equally important as hypothesis testing [28].

With $y_1, \ldots, y_n$ representing a random sample such that $y_i \sim Beta\ (\mu_i,\ \phi)$, the beta regression model is articulated as delineated in equation (2):

$$g\left(\mu_i\right) = x_i^T \beta = n_i \tag{2}$$

where, $\beta = (\beta_1, \ldots, \beta_k)^T$ represents a $k \times 1$ vector of unknown regression parameters ($k < n$), $x_i = (x_{i1}, \ldots, x_{ik})^T$ denotes the vector of $k$ explanatory variables and, the mean is presumed to be a linear function of the predictors [30]. Also, $g(\mu_i)$ is a link function $g: (0,1) \rightarrow R$, which is strictly monotonic and twice differentiable. Consequently, it is essential to acknowledge that beta regressions encompass two sub-models: one for the mean response and another one for the precision. As such, efficient parameter estimation in beta regression model depends on the correct modeling of the dispersion [31]. Cognizant of this requirement, it is important to model beta regression with varying and fixed dispersion and compare their performances. Along these lines, the merits of parsimony- where a model has to be selected as a subset of parameters gathered during the investigation, come into play. Model selection is crucial in regression analysis, as neglecting a significant predictor may result in biased outcomes, whilst the inclusion of irrelevant predictors can significantly impair estimation efficiency [32]. In this regard, three decisions during model selection are important, namely; i) assumption of the response

distribution; (ii) selection of the appropriate link functions, and (iii) the determination of which linear predictor(s) to include in the regressions [30]. The appropriate specification of the link function can be evaluated by the mis-specification test, whereas the predictors can be examined using goodness of fit tests like the Akaike Information Criterion (AIC) or visual assessments of model fits for smaller datasets [27].

Notwithstanding the predictive strengths, beta regression is limited under the frequentist framework where the response variable y ∈ [0,1] [33]. To estimate observations existing at 0 or 1, or ordered beta, an inflated beta regression under the Bayesian framework is applied with four parameters [34]. The inflated beta regression models encompass a mixture of models where values of 0 and 1 are modeled using the logistic regression while the values in between are modeled using the beta regression.

## Random forest regression

Within the machine learning framework, a random forest represents a specific category of ensemble learning algorithm [35] with numerous randomly generated trees, each casting a vote for a class [21]. Random forest (RF) regression consistently forecasts a continuous output by employing an ensemble of many decision trees [36]. The RF algorithm constructs a substantial ensemble of de-correlated decision trees by utilizing random selections of data and predictors, with each tree independently forecasting a result. The randomness in selecting training instances and input features introduces diversity among decision trees, facilitating the effective capture of relationships between input features and target variables [35]. This process de-correlates the trees, enhancing the robustness of the random forest against noisy data. In addition to effectively handling collinear data for final predictions, random forests aggregate the predictions of individual trees, thereby mitigating over-fitting and enhancing accuracy [37]. The strength of a RF process, is in the assessment of the importance between an input feature on the dependent variable, specifically its contribution to the overall model fit [38]. In ensemble approaches, variable importance is computed based on the mean square error reduction associated with it. An input feature that exerts influence on the output variable dwindles the model's predictive accuracy. Mean decrease accuracy as an indicator of variable importance demonstrates the extent to which the predictive accuracy reduces with the exclusion of a variable from the decision trees [36]. Nonetheless, the more reduction in the prediction accuracy, the stronger the relationship between the feature input and dependent variable. Furthermore, random forests possess the advantage that decision trees can be utilized for both regression and classification tasks [19].

Given that its slowly gaining traction in data science, its inherent strengths in reduced over-fitting risk, and enhancing predictive power by returning important variables based on reduction in mean square error, we tested the application of RF regression to have a picture of which important factors influence the dependent variable.

## Generalized additive models

Generalized additive models (GAMs) come in handy cognizant of the need to identify and characterize non-linear covariate effects. The linear predictor includes a sum of smooth functions of covariates, where the shape of the functions to be estimated is as shown in equation (3):

$$E\left(Y|X_1, X_2, \ldots, X_p\right) = g(E(Y)) = \alpha + s_1\left(X_1\right) + s_1\left(X_2\right) +, \ldots, s_p\left(X_p\right) \ . \tag{3}$$

where the $s_p$'s are the unspecified smooth nonparametric functions, and $g(E(Y))$ is the link function that links the expected value $E(Y)$ to the covariates. Each function is fitted using scatterplot smoother and then an algorithm for simultaneously estimating all $p$ functions is provided [20]. This flexibility implies that the predictive functions are unknown a priori but are uncovered by the modelling process of the underlying patterns in the data, thereby allowing an analyst to choose a fitting method appropriate for each explanatory variable. While other methods generally lack at least one of the three strengths, GAMs offer altogether, i) better interpretability, ii) flexibility-predictor functions can cover hidden patterns in data, and iii)

regularization-helps to reduce over-fitting [39]. From equation (3), it is important to note that smoothers (local regression, smoothing and regression splines) are the defining aspects of GAMs for they allow balancing of the bias-variance trade-off.

## Data, sources and management

Firstly, to obtain the dependent variable, this study constructed a composite indicator of service delivery using administrative data from the education, health and water sector for 136 districts/local governments in Uganda, including Kampala. The list of elementary indicators used to build the composite indicator is shown in *S1 Table*. Data points for the elementary indicators corresponding to the year 2021 were sourced from the Education Management Information System (EMIS), Annual Health Sector Performance Report-2022, Ministry of Water and Environment- Natural Resources, Environment, Climate Change, Land and Water Management Program Performance Report 2022, and the Uganda Bureau of Statistics website. For the process of CI building we refer to [40]. After a synthesis of literature on district local governments such as The Uganda Local Government Act 1997, Sector Service Delivery Assessments [11], third National Development Plan [41], and the Discretionary Development Equalization Grant (DDEG) Grant, Budget and Implementation Guidelines [42], we extracted potential predictors. This aligns with Chowdhury and Turin (2020), who posited that candidate variables for a specific topic can be selected based on subject matter knowledge or by reviewing the existing literature on the topic [43]. It is our position that these predictors would impact the performance of district local governments regarding service delivery. A brief description of the predictor variables is shown in Table 1. Data points for the predictors corresponding to the year 2021 were obtained from various sources such as budget dashboard [48], government finance statistics, and administrative geography file. These were already published data by mandated authorities. Because of deficiencies in data supply for all higher local government levels the data for the municipalities or cities were added to the district data for easier comparison. Descriptive statistics in form of measures of central tendency and of dispersion were performed to reveal the pattern and structure of the data [44]. This aided in identification and rectification of outliers and cases with missing values prior to bivariate and further analyses.

## Building of the composite indicator of service delivery

To obtain reliable differentials, it was important that a tested approach is adopted to construct a composite indicator of service delivery in line with the OECD recommendations [45]. Fig 1 is a flow chart detailing the general steps followed to build the composite indicator.

As shown in Fig 1, the construction of the CI started with the collation of data points for the selected elementary indicators by dimension. Exploratory data analysis was performed as a part of preliminary data analysis to determine whether a variable was symmetrical or not. In case of non-symmetry, the variable was winsored. Correlation analysis was performed as a data dimension reduction approach to signal variable redundancy within the dimension. Next, was the data normalization phase followed by first-stage aggregation to obtain the dimension sub-indices. The weighting of the sub-indices by testing the four methods was done in combination with either additive or multiplicative aggregation to obtain the CI scores that were ranked. Robustness tests using average shifts in ranks for the approach as well as the district local governments were performed, upon which the scores from the most stable approach were adopted as the values for the dependent variable.

## Theoretical framework for public service delivery assessment

In the building of composite indicators, a robust theoretical framework delineates the multi-dimensional phenomena under measurement, its dimensions, pillars, justifies the selection of elementary indicators, and, in certain instances, the weights to be applied [45]. This paradigm is derived from a combination of two theoretical perspectives: the World Bank's public administration production function [46] and the methodology for assessing public service delivery by Amin and Chaudhury

**Table 1. Description of the variables used in this study.**

| No. | Variable name | Description | Content | Units | Framework | Assumptions for inclusion |
|---|---|---|---|---|---|---|
| 1 | Popn_2021 | Population | The total number of people living in that district as per the Census methodology | Number | This is a parameter used in determining allocation of budgetary resources to local governments | Provides for the demand and pressure for services. More population may translate into more taxpayers for local revenue generation to finance work plans. |
| 2 | Land_area | Land area | The size is measured through the estimated number of square kilometers of land for a given district. This does not include land covered by water. | Sq km | Section 179 (4) of the Constitution | With low levels of infrastructure development, the size of operating area resonates with proximity to centre of decision making thus may influence effective monitoring for service provision, |
| 3 | Local_revenue | Local revenue mobilized | The amount of money generated by a local government through | Amount (UGX) | Article 191 of the Constitution, Section 80 of the LG Act 1997 | Districts retain a proportion of local revenue which may be used to provide public services |
| 4 | Sub_cty | Number of sub counties | A sub-county is a lower local government, usually in rural areas, below a district in administrative hierarchy. A summation of sub-counties makes up a district. | Number | Part 2 of the second schedule of the LG Act 1997 | Sub-counties have distinct mandates that must be financed separately. The number of sub-counties influences the available resources needed for service provision |
| 5 | Town_cl | Number of town councils | This is equivalent to a sub-county; however, it is urbanized/located in a town. | Number | Section 30(3) of the LG Act 1997 | Town councils are urbanized areas and have distinct mandates that must be financed separately from the district |
| 6 | Parish | Number of parishes | This is an administrative area below the sub-county. A group of neighboring parishes make up a sub-county | Number | Parish Development Model (PDM) | Parishes under the PDM are financed to plan and provide public services. More parishes may put pressure on the meagre resource envelop for services. |
| 7 | Centgovt_fund | Funding from the central government-consolidated fund | This is money allocated from the national treasury to local governments to execute their mandates under the national budget framework | Amount (UGX) | Part IV Section 74 of the LG Act, Budget and Implementation Guidelines | This is the major source of resources for providing services. More funding may be used to improve the situation. |
| 8 | Age_2022 | Age of the district | The number of years of existence of the district from the time it was created/gazette under the relevant laws | Years | Article 179 of the Constitution. | There has been an increase in creation of new districts under the rationale of "bringing services closer to the people" |
| 9 | Distance_kla | Distance of the district headquarters from the capital city | The number of kilometres, by road from Kampala to the district headquarters | km | Section 179 (4) of the Constitution | The second article showed that Kampala, the capital city, ranked top in provision of services, followed by the neighboring regions. Could it be an issue of longer to reach areas? |
| 10 | Donor_fund | Amount of donor funding | The money received by districts from development partners/non-government organizations to execute their mandates | Amount (UGX) | Public Finance Management Act, 2015 | Amount of donor funding implies the extent of extra resources for provision of services |
| 11 | Exp_3sectors | District expenditure on 3 sectors | The money spent by districts in a financial year is in the health, education and water sectors. | Amount (UGX) | Part 3 of the Public Finance Management Act, 2015 | The district may have high revenue but may not spend it on the 3 sectors considered under this study. Sections 20 and 22 of the Act provides for re-allocation of funds. |
| 12 | Is_prdp | Whether the district is PRDP (1-Yes, 2_No) | Peace Recovery and Development Plan (PRDP) for Northern Uganda is implemented in some districts as part of affirmative action for districts that were affected by conflict in the 1990s. | – | 2023 Discretional Development Equalization Grant (DDEG), Constitution Article 193 (4). | Effects of conflict in service provision may take time to be mitigated completely for such districts to move in tandem with other districts |

*(Continued)*

**Table 1.** (Continued)

| No. | Variable name | Description | Content | Units | Framework | Assumptions for inclusion |
|-----|---------------|-------------|---------|-------|-----------|---------------------------|
| 13 | Is_urban | Whether the district has a city or municipality (1-Yes, 2-No) | If district data was merged with a city or municipality. These are urbanized areas/towns. | – | Part 2 of the LG Act | Urbanisation may influence the provision of public services, through additional revenue and heightened demand concentrated in a small area. |
| 14 | Is_disaster | Whether the district is disaster-prone (1_Yes, 2-No) | A district is categorized as having high risk of experiencing disasters such as landslides and floods | – | DDEG allocation formula, National Risk and Vulnerability Atlas, 2019 | Environmental Disasters obstruct or destroy critical infrastructure for service provision |
| 15 | Is_refugee | Whether the district is a refugee hosting (1_Yes, 2-No) | A district that has a refugee camp located within its borders | | DDEG allocation formula, UNHCR | Refugee populations may put pressure on the available public services, or through extra support, more services may be initiated due to the presence of refugees |
| 16 | Centgovt_other | Funding from other central government entities | Money received by districts from other central government institutions | Amount (UGX) | Part 3 of the Public Finance Management Act | More funding leverages the ability of the district to provide more or better public services |

(2008) [47]. According to Amin and Chaudhury, improved measurement and evaluation of service delivery occurs along the intended results chain from the governance intervention to the quality of services. The World Bank's service production function highlights the importance of culture as a pivotal element around which service inputs and outcomes revolve. This is important to contextualize service performance measurements in terms of elementary indicators, as whatever is considered normal fluctuates by national settings and stage of development. Accordingly, we construe our theoretical framework as follows: Service delivery originates from triggers such as development frameworks, mandates, and policies that inform governance decisions made by policymakers at both central and local government levels. Consequently, experts in both state and non-state institutions execute these decisions by transforming service inputs into outputs provided to users at front-line service delivery points, including schools, health centers, roadways, and water points. Resultantly, through the contact between providers and seekers at delivery points, service-seekers cultivate an experience that manifests as an evaluation of the services' fitness for purpose. This framework conceptualizes service delivery as a results chain, illustrating the logical links of cause and effect from inputs to processes, outputs, and ultimately outcomes. Under this theoretical framework, i) elementary indicators assessing service delivery across all levels of the results chain were incorporated into the CI, classified under three sectors/dimensions, ii) the dimensions are mutually exclusive, facilitating the initial aggregation to derive sub-indices, iii) global and national service delivery standards were incorporated using distance-to-reference point data normalization, iv) the objective of service delivery is to optimize supply to satisfy demand; thus, the service-seeking behaviors and experiences of active recipients were taken into account through the selection of elementary indicators, and v) the elementary indicators selected largely reflect to the context of a developing country. Where available, the service delivery standards are presented as ideal/utopian values for the elementary indicators (see *S1 Table*).

The primary stages of composite indexing are data normalization, weighting, aggregation, and robustness testing. The identification of utopian values, which would ideally serve as global service standards, is connected to the initial phase (development framework and mandate) of service delivery within our proposed theoretical framework. The distance-to-reference point approach of data normalization used these utopian/idealistic values for data transformation and facilitates worldwide comparisons in service performance. This study evaluated four weighing methods classified as data-derived

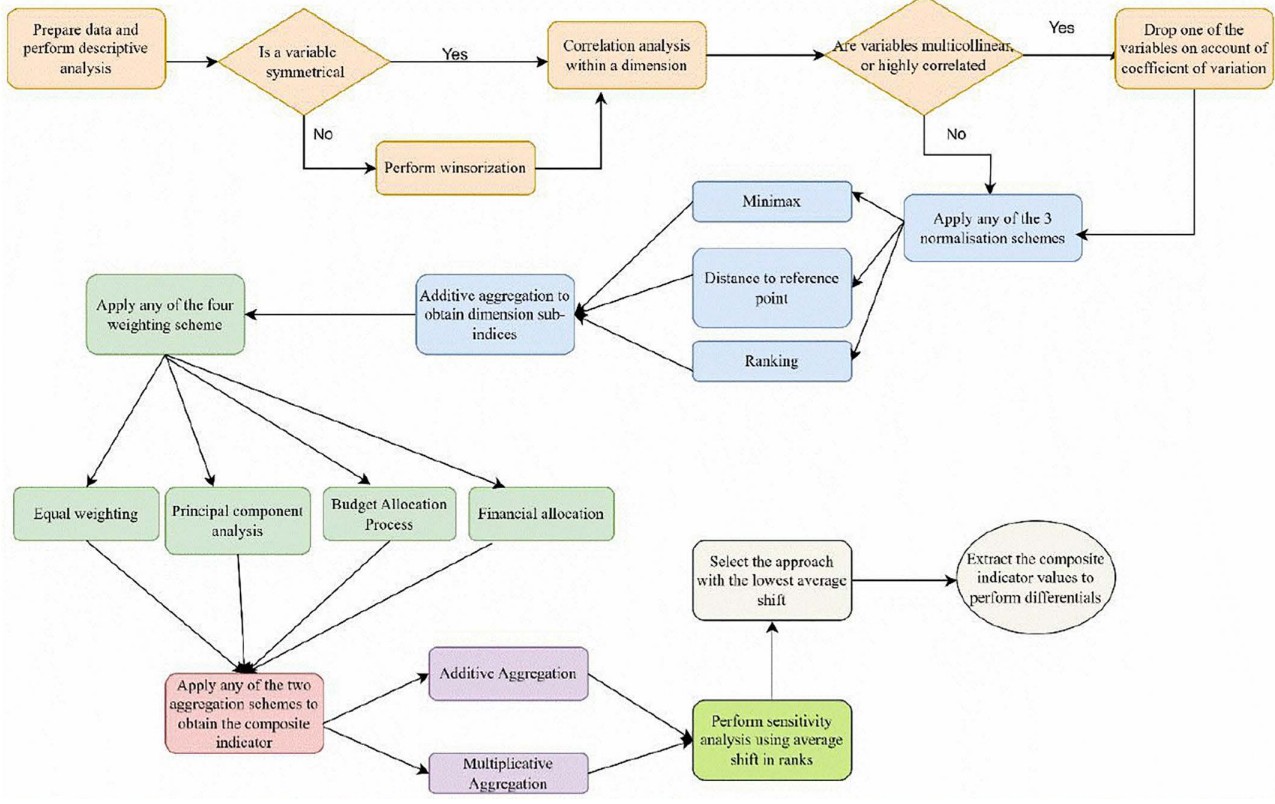

**Fig 1. Steps followed to construct a composite indicator of service delivery, as synthesized from OECD guidelines.**

and externally-derived. The data-derived weighting approaches include equal weights and principal component analysis, which are more empirical and typically yield robust CI rankings. Furthermore, the application of additive aggregation weights entails value judgments; thus, employing equal weights signifies equal significance of the elementary indicators within the dimension, as well as among the dimensions that constitute the overall CI. This is an important concern in contexts where service centers acknowledge all sectors as equally significant to service recipients, so expressing equality as a service objective. The externally sourced weights acknowledge the significance of context in influencing service delivery results. The financial analysis weights were determined by the proportion of the national budget assigned to each of the three sectors, acknowledging that the assessment of service performance should fluctuate with the level of financial inputs, as funding, *ceteris paribus*, is a significant factor in enhancing service outputs. The budget allocation weights were derived from experts (local government planning officers) to indicate the areas they deem crucial for enhanced service performance. This participatory approach facilitates the gathering of insights from specialists in the CI construction process [49] while ensuring equitable acknowledgment of service contributions.

## Exploratory data analysis

We categorized the 20 elementary indicators according to three dimensions (see *S1 Table*) and thereafter analyzed the corresponding variables for descriptive statistics- mean, median, variance, skewness, kurtosis, and coefficient of variation. Variables that were asymmetrical (skewness>2 or kurtosis>3) were winsored. The Pearson correlation coefficient between any two variables within the dimension was obtained to guide in data dimension reduction using a threshold of less than

−0.3 for highly negative, above 0.7 for highly positive correlation. Only two variables surpassed the thresholds, leading to the exclusion of one; thus, 19 variables were incorporated into the final construction model. Three dimensions- education, health and water- were selected for assessment because according to part two of the second schedule of the Uganda Local Government Act, 1997 (as amended) the three are the topmost mandated service sectors under the jurisdiction of the district local governments [13]. Moreover, the three sectors consume over 20% of the national budget [48]; thus, it is logical for service delivery assessments to prioritize outcomes in these sectors. Nonetheless, the composite indicator would have been enhanced by the incorporation of additional recognized service sectors, including justice, transportation, and electricity, along with a varied array of elementary indicators that thoroughly address service quality, quantity, infrastructure, access, and equity; however, this was constrained by the lack of district-level data for these themes and sectors.

### Data normalization stage

The selected variables were normalized at the subsequent stage using any of the three methods- ranking, minimax or distance to reference point. Minimax normalization involves subtraction of the value from the minimum and dividing by the variable range. This method does not reshape the distribution of the data. Distance to reference point involves dividing the value with the reference value/point. For some variables, the reference point was obtained exogenously from service delivery standards or development framework targets, while for the rest it was the maximum value possible for that variable. The ranking method transforms the data into ranks by taking the position of the value relative to the highest performing value for that variable. This method reshapes the distribution of the variable. These three methods were applied as plausible scenarios in composite indexing to aid in testing the robustness of the composite indicator, but also to offer basis for comparison and selection of the most reliable combination of approaches.

### Aggregation

At the first stage of aggregation, normalized variables were aggregated additively and with equal weighting within each dimension to obtain dimension sub-indices. The additive aggregation at this stage enabled compatibility of the values obtained through minimax and distance to reference point with multiplicative aggregation at the second stage.

At the second stage, dimension sub-indices with the independent application of four weighting schemes (equal weighting, principal component analysis, financial allocation, and budget allocation process) were aggregated additively and multiplicatively. Twenty-four sets of composite indicator scores and associated ranks for each district local government were obtained through a combination of methods as illustrated in the Table 2. The ranks allowed for the computation of the average shifts in ranks as a test of uncertainty/ robustness. The results showed that the constructed composite indicator is stable, but also provided an opportunity to compare the 24 sets of approaches to enable the selection of the most robust. The set that produced the lowest average absolute shift in ranks (minimax transformation- equal weighting-multiplicative aggregation) was considered the most stable, and therefore its scores were adopted as the composite indicator of service delivery for district local governments in Uganda [49]. It is this composite indicator scores that were used as the outcome variable for this study.

To determine the relationship between the outcome variable (CI) and the predictors, that is the variables that could explain the service delivery differentials in Ugandan local governments the three statistical approaches narrated in the preceding section – Beta, random forest regression and generalized additive model – were applied.

Some measures of central tendency and dispersion were performed for each of the independent variables to show patterns in the performance of the district local governments as shown in Table 2. Missing cases were identified and addressed by assigning median values for that variable. Outliers were determined using box plots and addressed in two variables (population size and central government funding selected arbitrarily) using the inter-quartile range method leading to a 124X17 dataset. Two-way tables were produced for categorical variables in addition to histograms for the distribution of continuous variables.

**Table 2. Sets of methods leading to the composite indicator scores.**

| No. | Combination of methods at normalization, weighting and aggregation | Average shift in Rank ($\overline{R_s}$) |
|---|---|---|
| 1. | Minimax-equal weighting-additive | 7.23 |
| 2. | Minimax-principal component analysis-additive | 7.53 |
| 3. | Minimax-financial allocation-additive | 6.99 |
| 4. | Minimax-budget allocation-additive | 7.00 |
| 5. | Minimax-equal weighting-multiplicative | **6.22\*** |
| 6. | Minimax-principal component analysis-multiplicative | 6.24 |
| 7. | Minimax-financial allocation-multiplicative | 8.19 |
| 8. | Minimax-budget allocation-multiplicative | 10.46 |
| 9. | Distance to reference point-equal weighting-additive | 16.72 |
| 10. | Distance to reference point -principal component analysis-additive | 19.80 |
| 11. | Distance to reference point -financial allocation-additive | 18.86 |
| 12. | Distance to reference point -budget allocation-additive | 17.06 |
| 13. | Distance to reference point -equal weighting-multiplicative | 15.4 |
| 14. | Distance to reference point -principal component analysis-multiplicative | 17.66 |
| 15. | Distance to reference point -financial allocation-multiplicative | 18.20 |
| 16. | Distance to reference point -budget allocation-multiplicative | 17.23 |
| 17. | Ranking-equal weighting-additive | 8.72 |
| 18. | Ranking -principal component analysis-additive | 9.74 |
| 19. | Ranking -financial allocation-additive | 9.36 |
| 20. | Ranking -budget allocation-additive | 9.09 |
| 21. | Ranking -equal weighting-multiplicative | 9.26 |
| 22. | Ranking -principal component analysis-multiplicative | 10.27 |
| 23. | Ranking -financial allocation-multiplicative | 9.83 |
| 24. | Ranking -budget allocation-multiplicative | 8.988 |

*\*The lowest absolute shift in ranks implies the most stable approach.*

Considering that the composite indicator scores are akin to percentages in the range, $y \in (0,1)$, beta regression, as reviewed from literature, is the appropriate statistical approach as shown by equation (2). We began by running the full model with all the sixteen predictors a fixed precision parameter and equally the same model but with a varying precision parameter in respect to some continuous variables. On the basis of their Information Criteria (IC), one model was selected however, this further necessitated regularization using step Beta function to select the important variables in pursuit of parsimony. The selected approach was used to explain the differentials in service delivery aided by the level of significance. The coefficients were corrected for bias in pursuit of efficient estimates [50]. Model diagnostics using tools such as Cook distance, standardized weighted residual plot, half-normal plots and local influence method were performed through residual plots. We tested the predictive accuracy of the selected model by learning from it with a training (70% of the observations) and testing datasets.

To perform the random forest algorithm, five (corresponding to $m_{try} = p/3$) randomly selected variables were used to split each node from the bootstrapped datasets to generate a decision tree in a random forest training. The minimum node size was 5. Mean decrease accuracy (MDA), was used to evaluate the importance of each variable in its impact on predicting service delivery in Ugandan local governments. A comparison of the fitted values for random forest regression and Beta regression regularized model was performed and it showed significant correlation. Whereas, the out-of-bag (OOB) error rate can conclusively be used to evaluate the predictive accuracy of a RF model with especially small sample sizes as it

uses bootstrapping, in this study we as well used the training and testing datasets upon which 500 decision trees were built and an ensemble of results aggregated. This was intended to provide a consistent procedure for testing the predictive accuracy applied to the Beta regression and GAM in this study.

The dataset was subjected to a GA model through specification of the smooth functions for the continuous variables. Model regularization was performed in pursuit of parsimony. The selected variables/model were run against the training and testing dataset to evaluate the predictive accuracy. The functional forms for the covariates were observed through the plots and provided basis for reporting their trajectory and relationship with service delivery at local government level.

## Results

The distribution of the composite indicator is as shown by the bee swarm and density plot in Fig 2. Performance varies from a score of 0.25 to less than 0.60 on a scale of (0,1). The extensive region of the density plots beneath 0.50 indicates that numerous districts scored below the 50% threshold, signifying a marginally acceptable level of performance in service delivery.

The density plot in Fig 2 is nearly symmetrical, albeit with negligible tendencies of negative skewness, caused by infrequent very low scores, probably for two districts. A distribution is called a bimodal probability distribution, if $f_x(x)$ has exactly two maxima. Whereas the density plot shows two peaks, it is largely unimodal. The small peak on the left is caused by a small number of outliers. Generally, the plot suggests that the service delivery CI scores across districts, lack clusters or subgroups. Therefore, it is reasonable not to treat subsequent analysis as classifications. The plot exhibits a degree of breadth, suggesting modest variability in the composite indicator scores. The tail on the left signifies an increased likelihood of extreme values (districts with scores significantly divergent from others). The histogram shows that composite indicator values (if represented by random response variable $y$) run from 0.2 to 0.6, therefore $y \in (0,1)$ imply an absence of zero or one inflated values. Subsequent regression modelling may be incognizant of inflated values.

In terms of descriptive statistics as illustrated in Table 3, there are high variations among the districts in population, number of town-councils, and all the financial indicators. As initial tests of symmetry, skewness ($|\acute{\alpha}3| > 2$) and kurtosis ($\acute{\alpha}4 > 3$) indicates asymmetrical (high variations) data. The high variances indicated by kurtosis are found in the levels of donor money to districts, highlighting extreme values of donor assistance to certain districts. All the predictor variables exhibit positive skewness (mode<median<mean), indicating that most districts have higher frequency of low predictor values. Generally, the districts in Uganda largely vary in funding, population, and geographical size.

Variables exhibiting elevated kurtosis and skewness indicate the existence of outliers, necessitating remediation prior to their incorporation into the regression model. Outliers exhibit disproportionate influence, distorting the mean and estimates away from typical values, so rendering the model less representative of the underlying data, resulting in biased parameter estimates, incorrect confidence intervals, and inaccurate predictions. Therefore, modelling the composite indicator scores against such variables in their untreated form may be statistically detrimental.

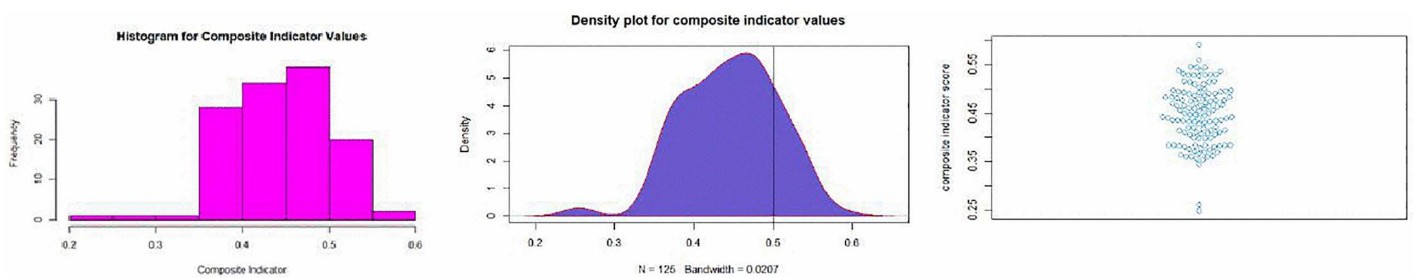

**Fig 2. Distribution of the service delivery composite indicator by district local government, 2021.**

**Table 3. Descriptive summaries for the continuous variables.**

| Variable | Mean | Median | Standard deviation | Skewness | Kurtosis | Coefficient of variation | Coefficient of range |
|---|---|---|---|---|---|---|---|
| popn_2021 | 323548.53 | 259700.00 | 312612.33 | **6.05** | **48.79*** | 0.97 | 0.97 |
| land_area | 1513.37 | 1206.70 | 1089.08 | 1.40 | 1.88 | 0.72 | 0.95 |
| sub_cty | 11.15 | 10.00 | 5.58 | 1.37 | 2.18 | 0.50 | 0.78 |
| town_cl | 5.06 | 4.00 | 2.95 | 1.73 | **5.04*** | 0.58 | 0.90 |
| parish | 63.76 | 56.00 | 33.61 | 1.66 | **4.15*** | 0.53 | 0.88 |
| age_2022 | 22.82 | 16.00 | 18.37 | 0.89 | −0.59 | 0.81 | 0.97 |
| distance_kla | 276.81 | 278.00 | 132.03 | 0.25 | −0.25 | 0.48 | 1.00 |
| local_revenue | 54.83 | 32.65 | 76.37 | **5.37** | **39.13*** | **1.39** | 0.99 |
| Centgovt_fund | 3175.41 | 2583.82 | 4334.49 | **10.33** | **115.12*** | **1.37** | 0.99 |
| Centgovt_other | 258.85 | 87.56 | 512.57 | **3.17** | **9.42*** | **1.98** | **1.00** |
| donor_fund | 129.74 | 38.90 | 876.59 | **11.59** | **134.81*** | **6.76** | **1.00** |
| exp_3sectors | 1872.94 | 1730.29 | 934.35 | **2.55** | **11.37*** | 0.50 | 0.87 |
| composite | 0.45 | 0.45 | 0.06 | −0.17 | 0.31 | 0.14 | 0.43 |

*Leptokurtic variable (kurtosis ά4 > 3)

Table 4 presents the frequencies for categorical variables, revealing that 28% of the districts are urbanized (including a city or municipality), 32% are susceptible to catastrophes or have a high risk of environmental calamities, and 9% are designated as refugee-hosting districts. According to the United Nations High Commissioner for Refugees (UNHCR), Uganda accommodates the largest population of refugees and those requiring international protection in Africa, totaling over 1.7 million, and ranks fifth globally. The refugee population is assumed to augment the demand for services offered by local governments. The two-way frequencies of the categorical variables are also displayed in S2 Table, indicating that only 10% of disaster-prone districts are urbanized, 8% are designated as Peace Recovery and Development Plan (PRDP), and 30% accommodate refugees. Furthermore, of the PRDP districts, 6% are urbanized, whilst 26% accommodate refugees.

Urban local governments exhibit a significant concentration of employed, highly educated individuals within a limited geographical scope, facilitating the informed demand and heightened provision of services in contrast to rural communities. Disaster-prone districts frequently experience natural calamities that devastate essential service infrastructure, including roads, water sources, and educational facilities. This destruction requires substantial rehabilitation funding, which is often not readily accessible, to restore service delivery to the standards of other local governments. The PRDP and local governments in refugee-hosting districts obtain supplementary funds from the central government to improve service delivery; thus, explicitly distinguishing them in the study may yield insights into the effects of these specific funding contributions.

The correlation matrix in S3 Table reveals low bi-variate associations among most of the continuous variables, except for the correlation between central government funding and district expenditure across the three sectors ($\rho = 0.86$). This suggests that district allocations for these sectors predominantly depend on central government budgetary allocations.

**Table 4. Frequency distributions for categorical variables.**

| | is_urban | is_disaster | is_prdp | is_refugee |
|---|---|---|---|---|
| **Yes** | 0.28 | 0.32 | 0.26 | 0.09 |
| **No** | 0.72 | 0.68 | 0.74 | 0.91 |
| **Total** | 100 | 100 | 100 | 100 |

Unexpectedly, there exists a low and negative correlation between donor support and expenditure across three sectors ($\rho = 0.03$), suggesting that a significant portion of donor funding to districts may not be allocated to these three essential service delivery sectors, or pertains to different components than those utilized to construct the composite indicator.

### Beta regression model

The results of the Beta regression model with fixed and varying dispersion parameter are shown in Table 5 below. The model with fixed dispersion parameter generates four significant predictors that are the number of sub-counties, number of town councils, distance of the district headquarters from the capital city and land area. The model with varying dispersion parameter generates five significant predictors for the mean model namely, number of sub-counties, number of town

**Table 5. Results of application of Beta regression.**

| Model with fixed phi | | | | Model with varying phi | | |
|---|---|---|---|---|---|---|
| | Mean model with logit link | | | Mean mode with logit link | | |
| Variable | Estimate | SE | p-value | Estimate | SE | p-value |
| Intercept | −0.2339 | 0.0924 | 0.0114* | −0.2852 | 0.0879 | 0.0012** |
| Population | −0.0000 | 0.0000 | 0.2037 | −0.0000 | 0.0000 | 0.1797 |
| Land area | −0.0000 | 0.0000 | 0.0571. | −0.0000 | 0.0053 | 0.0469* |
| # sub-counties | −0.0165 | 0.0053 | 0.0019** | −0.0225 | 0.0102 | 0.0000*** |
| # town councils | 0.0277 | 0.0115 | 0.0158* | 0.0208 | 0.0008 | 0.0415* |
| # parishes | 0.0011 | 0.0009 | 0.2616 | 0.0007 | 0.0014 | 0.3885 |
| Age of the district | 0.0009 | 0.0777 | 0.5848 | 0.0014 | 0.0002 | 0.3032 |
| Distance from the city | −0.0004 | 0.0002 | 0.0370* | −0.0312 | 0.0592 | 0.1417 |
| Urbanized district | −0.0235 | 0.0777 | 0.7620 | −0.0311 | 0.0592 | 0.5908 |
| Disaster prone | −0.0606 | 0.0458 | 0.1863 | −0.0311 | 0.0426 | 0.4649 |
| PRDP district | 0.0481 | 0.0554 | 0.3847 | 0.0575 | 0.0465 | 0.2169 |
| Refugee hosting | 0.0971 | 0.1572 | 0.5369 | −0.0347 | 0.1066 | 0.7443 |
| Local revenue mobilized | 0.0005 | 0.0004 | 0.8251 | 0.0000 | 0.0003 | 0.7465 |
| Central Gov't funding | 0.0000 | 0.0000 | 0.2689 | 0.0001 | 0.0000 | 0.0044** |
| Other Central Gov't funding | 0.0000 | 0.0000 | 0.3895 | 0.0001 | 0.0000 | 0.0252* |
| Donor funding | −0.0001 | 0.0003 | 0.7284 | 0.0000 | 0.0003 | 0.9732 |
| Expenditure on 3 sectors | 0.0000 | 0.0000 | 0.6902 | −0.0000 | 0.0000 | 0.4681 |
| Phi coefficients | 98.72 | 12.47 | 0.0000*** | | | |
| Pseudo R²= 0 32 | | | | Pseudo R²=0.27 | | |
| AIC=−356.89 | | | | AIC=−357.81 | | |
| BIC=−306.13 | | | | BIC=−290.12 | | |
| DF=24 | | | | DF=24 | | |
| **Precision model with log link Phi coefficients** | | | | | | |
| Intercept | | | | 2.7548 | 0.5764 | 0.0000*** |
| # sub-counties | | | | −0.0531 | 0.0281 | 0.0585. |
| # town councils | | | | −0.1057 | 0.0669 | 0.1141 |
| Distance from the city | | | | 0.0033 | 0.0011 | 0.0015** |
| Local revenue mobilized | | | | 0.0056 | 0.0031 | 0.0645. |
| Central Gov't funding | | | | 0.0011 | 0.0003 | 0.0002*** |
| Expenditure on 3 sectors | | | | −0.0006 | 0.0004 | 0.1597 |

Significance codes ***0.001, **0.01, *0.05, "."0.1

councils, land area, central government and other funding from central government agencies. For the dispersion model, four significant factors are generated namely; the number of sub-counties, central government funding, local revenue mobilized, and distance of the district headquarters from the capital city. The signs of the coefficients are similar for the mean models in both cases. The Akaike Information Criterion (AIC), Bayesian Information Criterion (BIC) and Pseudo $R^2$ show no significant differences in fit for the two models.

Variable selection is crucial as it enhances estimation accuracy by effectively identifying a subset of significant predictors, resulting in a parsimonious representation [32]. In terms of variable selection as shown in Table 6, the model with a fixed dispersion parameter identifies five variables: the number of sub-counties, the number of town councils, the distance of the district from the capital city, additional central government funding, and expenditure across the three sectors. The model with varying dispersion parameter identifies six variables (four are like the fixed dispersion model) namely- number of sub-counties, number of town councils, land area, age of the district, central government funding to the district and other central government funding.

The differences are rooted in the methodological approaches of these selection methods. Stepwise regression employs information criterion as shown in equation (4) to penalize the number of predictors included in the stepwise regression so as to maximize model fit with the fewest predictors [51]. AIC and BIC have different penalty levels, for AIC has a fixed penalty of 2, while BIC's penalty level- $ln(N)$, that increases with the sample size.

$$IC = -2 \log L(\hat{\theta}) + penalty\ (p) \tag{4}$$

**Table 6. Variables selected through model regularization.**

| Penalty | Beta Step | | | | | |
|---|---|---|---|---|---|---|
| | Model with fixed dispersion parameter | | | Model with varying dispersion parameter | | |
| Variable | Estimate | SE | p-value | Estimate | SE | p-value |
| Intercept | −0.2544 | 0.0818 | 0.0018*** | −0.3012 | 0.0683 | 0.0000*** |
| Land area | – | – | – | −0.0001 | 0.0000 | 0.0275* |
| # sub-counties | −0.0169 | 0.0043 | 0.0000*** | −0.0215 | 0.0044 | 0.0000*** |
| # town councils | 0.0224 | 0.0102 | 0.0279* | 0.0114 | 0.0078 | 0.1459 |
| Age of the district | – | – | – | 0.0017 | 0.0009 | 0.0716. |
| Distance from the city | −0.0004 | 0.0002 | 0.0279* | – | – | – |
| Central Gov't funding | – | – | – | 0.0001 | 0.00002 | 0.0000*** |
| Other Central Gov't funding | 0.0001 | 0.0001 | 0.0943. | 0.0001 | 0.00003 | 0.0101* |
| Expenditure on 3 sectors | 0.0002 | 0.0000 | 0.0036** | – | – | – |
| Phi | 91.46 | 11.5 | 0.0000*** | | | |
| Pseudo $R^2$ = 0.2661 | | | | $R^2$ = 0.2374 | | |
| AIC = −369.4214 | | | | AIC = −376.3556 | | |
| BIC = −349.6795 | | | | BIC = −345.3325 | | |
| DF = 7 | | | | DF = 11 | | |
| | | | | Precision model with log link Phi coefficients | | |
| Intercept | | | | 2.4996 | 0.5624 | 0.0000*** |
| # sub-counties | | | | −0.0678 | 0.0263 | 0.0101* |
| Distance from Kampala | | | | 0.0035 | 0.0010 | 0.0008*** |
| Central Gov't funding | | | | 0.0007 | 0.0002 | 0.0000*** |

Significance codes ***0.001, **0.01, *0.05, "."0.1

where *penalty(p)* quantifies the penalty assigned to the model incorporating *p* parameters. This quantity may be interpreted as a measure of divergence between the distribution of future data generated by random variable *Y* and that predicted by the model. The choice of the penalty identifies the type of information criterion where for AIC it is $-2\log(y; \hat{\theta})$ + 2p while for BIC it is $-2\log(y; \hat{\theta})$+p log n.

The $R^2$ is one of the metrics used to evaluate regression models, as it offers insights into the proportion of variability in the data explained by the model. By comparing Tables 5 and 6, the $R^2$ is low and reduces with model regularization, which is its typical limitation in that it increases with the number of predictors even if they do not necessarily contribute meaningful improvements in the model. The results show the parsimonious model obtained a pseudo-$R^2$ value of 0.237 which is comparable to Cribari-Neto (2023) who in his study on beta regression analysis of COVID-19 mortality rates in Brazil, obtained a pseudo-$R^2$ value of 0.347 [52]. Nonetheless, he utilized the residual half normal plot to test for beta-regression model fitness. This is reasonable, because the $R^2$ does not pass as the absolute panacea for model fitness/quality as it is sensitive to model complexity (over-fitting and under-fitting), and ignores residual analyses. To this end, other fit statistics such as information criteria, predictive abilities [53,54] and visual diagnostic plots [28] which may be applied singularly or in combination are invaluable for understanding model fitness/quality.

Consequently, we based goodness of fit evaluation on the AIC index, and we choose the model characterized by varying dispersion and regularized via stepwise beta regression. To obtain efficient coefficients in beta regression, bias correction or reduction is applied. The coefficients for the selected model are shown in Table 7.

From Table 7 and by using bias corrected coefficients, the number of sub-counties and land area have a negative relationship while Central Government funding from the treasury, funding from other agencies, number of town councils, age of the district have a positive relationship with service delivery. The intercept shows that when all predictors are zero the odds of service delivery outcomes (index) are reduced by about 27%. This implies that regardless of inclusion of the identified predictors, the service performance effect is below by 27%, suggesting the need for identification of more predictors to include in the model. The creation of an additional lower administrative units in the form of sub-counties, and keeping other predictors constant, reduces the odds of service delivery index by 3%. This could be ascribed to the increased cost of administering the sub-counties in the midst of low budgetary allocations. The financing for service delivery is diverted to cover running costs such as wages for staff, utilities, office space, travels, etc. The creation of an additional town council while keeping other factors constant increases the odds of service delivery outcomes (index) by 1%. Town councils are

**Table 7. Results of final model's efficient parameter coefficients chosen after correction of bias.**

| | Maximum Likelihood | Bias correction | Bias Reduction | Odds Ratio | p-value |
|---|---|---|---|---|---|
| **Mean Model** | | | | | |
| Intercept | −3.0122e-01 | −3.0107e-01 | −2.9836e-01 | 0.7399 | 0.0000*** |
| sub-counties | −2.1532e-02 | −2.1532e-02 | −2.1767e-02 | 0.9787 | 0.0000*** |
| Central Gov't funding | 9.0418e-05 | 9.0427e-05 | 9.2072e-05 | 1.0000 | 0.0000*** |
| Land area | −3.0896e-05 | −3.0863e-05 | −3.12006-e-05 | 1.0000 | 0.0275* |
| Other central gov't funding | 8.5315e-05 | 8.5236e-05 | 8.3646e-05 | 1.0001 | 0.0101* |
| Age of the district | 1.6987e-03 | 1.6973e-03 | 1.5243e-03 | 1.0017 | 0.0716. |
| # town councils | 1.1427e-02 | 1.1411e-02 | 1.1603e-02 | 1.0115 | 0.1459 |
| **Dispersion Model** | | | | | |
| Intercept | 2.499611e+00 | 2.609075e+00 | 2.707269e+00 | | 0.0000*** |
| # sub-counties | −6.781049e-02 | −6.761923e-02 | −6.598752e-02 | | 0.0101* |
| Distance from the capital city | 3.471522e-03 | 3.164311e-03 | 2.993898e-03 | | 0.0008*** |
| Central Gov't funding | 7.260960e-04 | 6.821994e-04 | 6.537176e-04 | | 0.0000*** |

Significance codes ***0.001, **0.01, *0.05

urban areas with higher population density making it easier for Government to provide services in a smaller geographical scope. Additionally, town dwellers are capable of mobilizing voices to demand for services than the rural counterparts. A unit increase in land size of a district reduces the odds of service delivery outcomes by 1%. Increased land area implies increased costs of service delivery supervision and monitoring, which may not be effectively realised. Predictors such as central government funding, other central government transfers and age of the district, though positive have very low odds ratios suggesting that a unit increase in their quantities changes the odds of service delivery outcomes by less than 1%. Of course, the funding element is an important input in service delivery especially for start-up of service points such as schools in areas where they do not exist.

The dual structure of beta regression modelling facilitates intricate analysis of data exhibiting heteroscedasticity, wherein both the mean and variability of the response are influenced by predictors. The precision model predicts the variability or precision of the response (service delivery composite indicator) utilizing logarithmic link. Two predictors- distance from the capital city and Central Government funding – have a positive relationship with the precision parameter while the number of town councils has a negative relationship with the precision parameter. A unit increase in the number of sub-counties decreases the log of the precision parameter by 0.006 units, thereby increasing dispersion (increasing variability) in the service delivery composite indicator. Conversely a unit increase in the distance of the district from the capital city increases the log of the precision parameter by 0.003 units, reducing dispersion (reducing variability) in the service delivery composite indicator. The central government funding has increased log of the precision parameter thereby reducing variability in the service delivery composite indicator.

Statistical analysis typically utilizes methods to assess whether a fitted model accurately represents reality. Accordingly, diagnostic tools utilizing residuals or influence metrics may be employed for that purpose [55]. The goodness of fit is assessed using different types of diagnostic displays shown in Fig 3a-3d.

Fig 3a and 3d employ standardized weighted residuals to account for leverages of the different observations and show observation randomly scattered, signaling absence of unusual features in the data which is a first indication that the Beta regression model with varying dispersion provides an appropriate representation for the data. From Cook's distance plot in Fig 3b, it is evident the stability of the model and its coefficients. Observation 105 is characterized by a significant Cook's distance and substantial residual. Therefore, to improve parameter accuracy, it may be excluded from the model in future studies. By employing the deviance, the plots in Fig 4a-4d further confirm the non-presence of substantial influential observations as shown by the half normal plot of residuals. Most of the residuals vary between 0 and 1.5; yet they all remain within the confidence bands as shown in Fig 4c, indicating a lack of definitive evidence for mis-specification of the Beta regression model.

In summary, the beta regression model with varying dispersion parameter has a proper fit for the data used to model the predictors of service delivery.

## Selection of important variables using random forest regression

A random forest regression model was run using the composite score as the outcome variable and five as the number of variables tried at each split. 500 hundred trees were grown, the mean of squared residuals at 0.0029 and the percentage of variance explained at 20%. The error rate is minimal and stabilizes from about 50 trees grown as shown in *S1 Fig*, an indication that growing about 50 trees could have as well yielded similar results.

RF regression assesses the importance of the predictors regarding the contribution to model accuracy. Higher importance means more influence on predictions. According to Table 8 and *S2 Fig*, the top five important variables in explaining service delivery composite indicator are: number of sub-counties with a percentage increase in mean square error at 12%, followed by central government funding, district expenditure on the three sectors, the district population and whether the district is a classified as a high disaster-risk.

Accessed literature suggests that when the increase in mean squared error (MSE) and the increase in node purity yield slightly divergent outcomes as seen in Table 8, the researcher should prioritize the results associated with the increase in

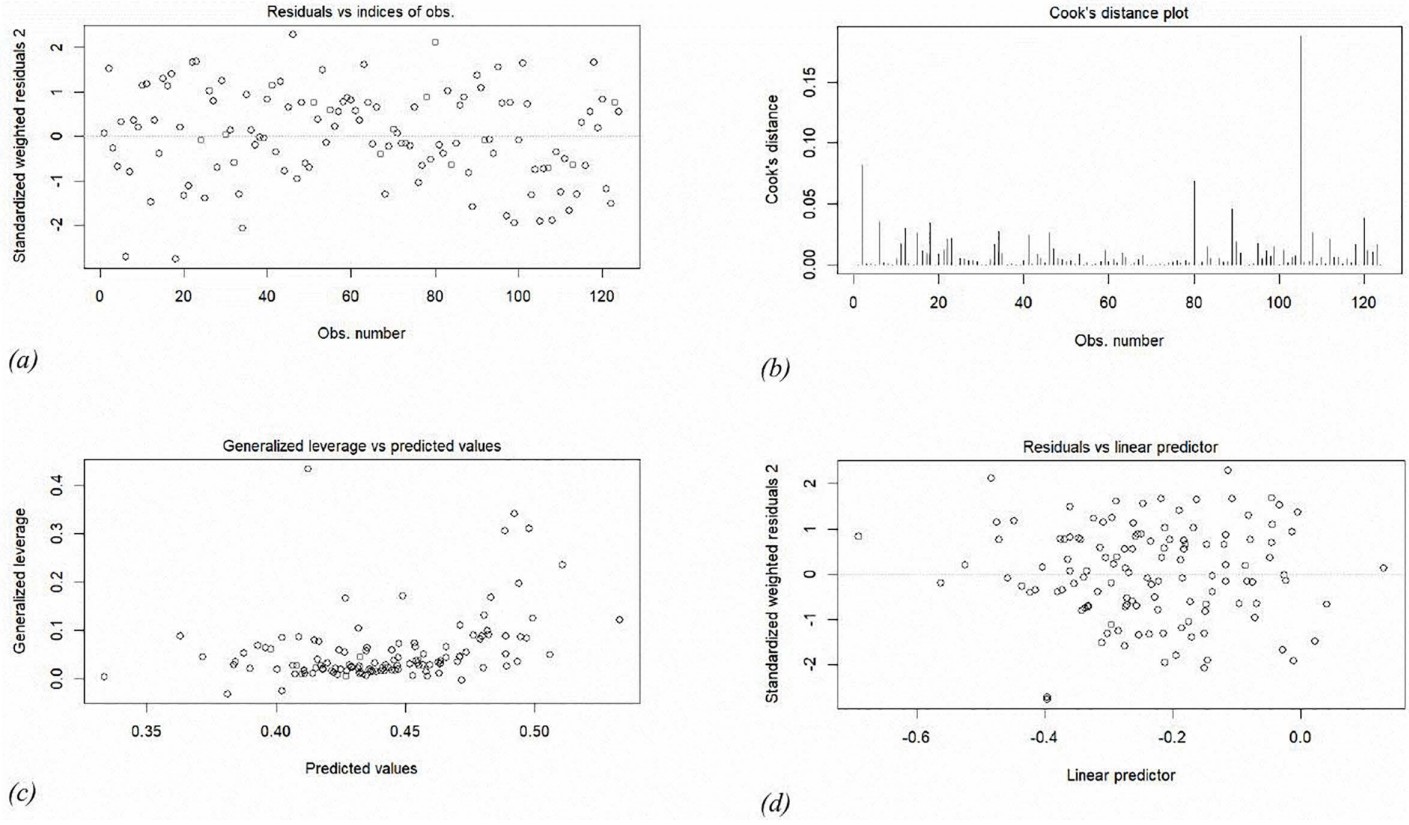

**Fig 3. Diagnostic plots for beta regression of service delivery composite indicator.**

MSE. This is because MSE directly relates to a model's predictive accuracy thus offering a more direct interpretation than increase in node purity.

A plot followed by correlation analysis of the fitted variables from the Beta regression model and predicted values of the random forest regression was significant at 0.05% level ($\rho = 0.69$, p-value = 0.00). Fig 5 shows a positive linear relationship hence suggesting that both approaches to some extent yield similar results, thus meriting their applicability in modelling service delivery in Ugandan local governments.

## Generalized additive model outputs

The coefficients of the GAM, after variable selection, are shown in *S4 Table* presenting significant effects for the predictors of number of town council, sub-county and central government funding. Notwithstanding the results and due to the underlying objective of GAM, it is often advised to make conclusions based on the plots.

Looking at the coefficients, sub-county and central government funding are the basis function weights that control the shape of these splines. For enhanced interpretability, these coefficients require a full understanding of what these basis functions look like and how they act collectively to form the spline, hence justifying the visualizing of the output using partial effects plots (Fig 6).

The plots in Fig 6 are necessary to visualize the individual predictor effects (one predictor when all other remaining predictors are dropped). The smooth plots for Central Government funding and number of sub-counties are slightly wiggly, which is uncovered through the GAM. Further, the diagnostic plots in *S3 Fig* confirmed that it is a good fit for this data.

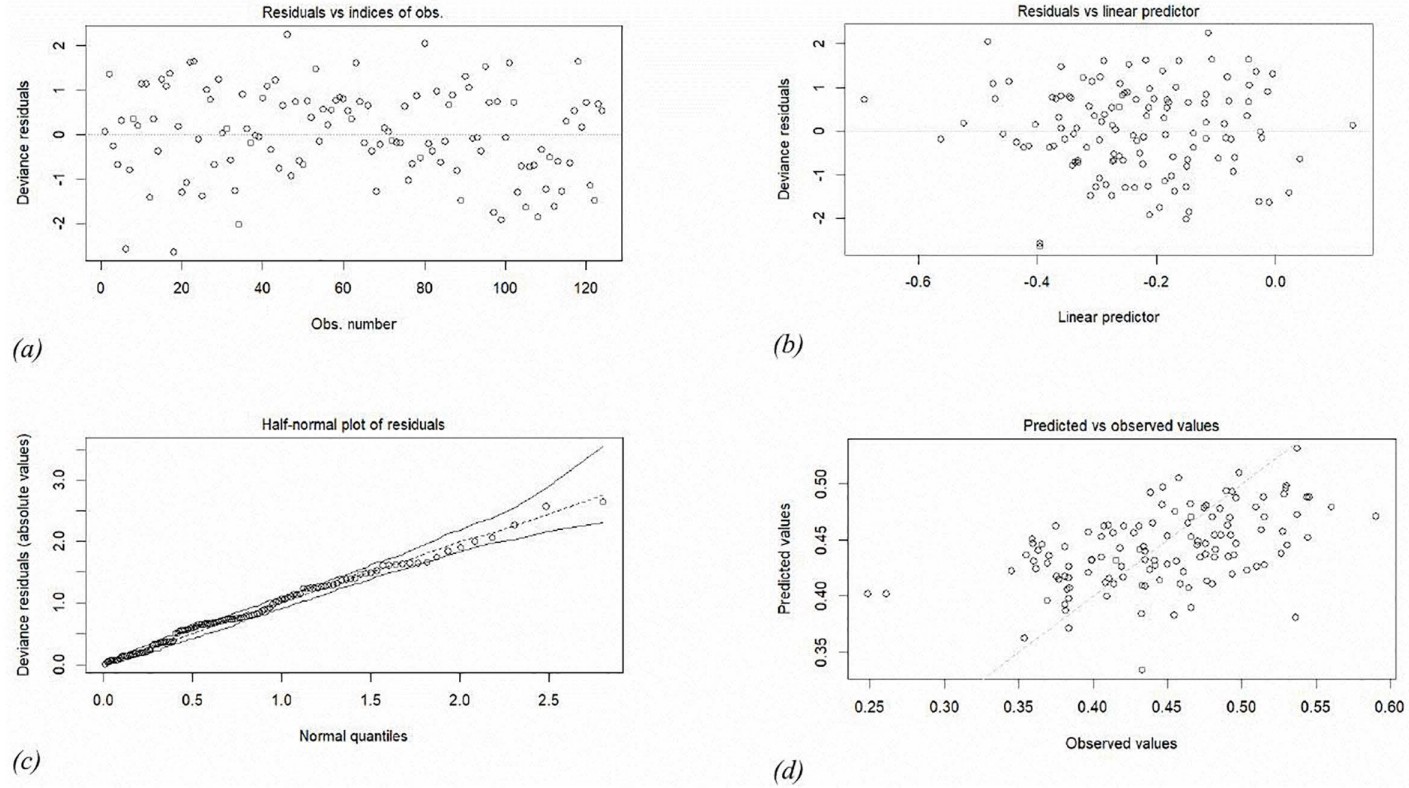

**Fig 4. Diagnostic plots for the beta regression model based on deviance tests.**

Table 8. Variable importance as returned by random forest regression.

| Variable | %Increase in MSE | Increase in node purity |
|---|---|---|
| Population | 6.6188 | 0.0359 |
| # sub-counties | 12.8684 | 0.0521 |
| Expenditure on 3 sectors | 8.4084 | 0.0468 |
| Central Gov't funding | 10.2386 | 0.0448 |
| Distance from the city | 3.0406 | 0.0422 |
| Local revenue mobilized | 2.1197 | 0.0415 |
| Land area | 0.3811 | 0.0326 |
| # parishes | 0.5396 | 0.0291 |
| Donor funding | −2.2386 | 0.0245 |
| Other Central Gov't funding | 0.5094 | 0.0235 |
| Age of the district | 0.2601 | 0.0172 |
| # town councils | 1.3866 | 0.0148 |
| Disaster prone | 5.8287 | 0.0086 |
| PRDP district | −0.3297 | 0.0022 |
| Urbanized district | 0.2942 | 0.0017 |
| Refugee hosting | 1.7088 | 0.0006 |

MSE = 0.0028 $R^2$ 0.2049

**Beta regression fitted values Vs RF predicted values**

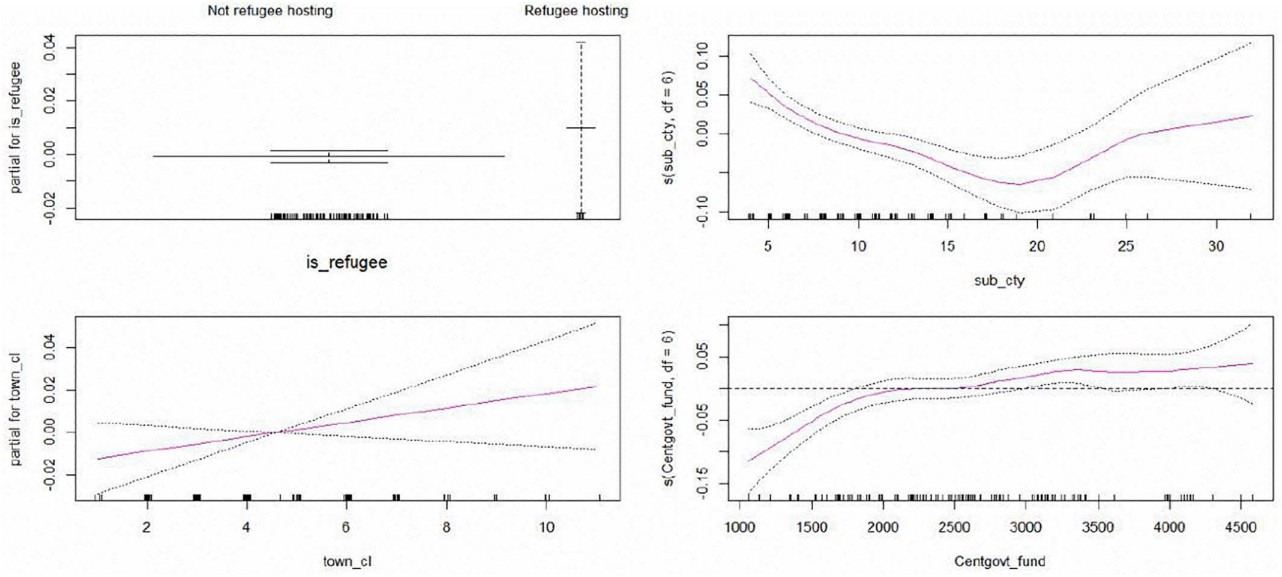

**Fig 5. Plot of the beta regression fitted values and random forest predicted values.**

**Fig 6. Plots from application of a generalized additive model.**

As shown in Fig 6, the plot for the Central Government funding confirms that the effect of this predictor is slightly negative for lower levels of funding but quickly becomes positive (above zero) for intermediate amounts of funding (about UGX 2.5 billion). It increases gently and then plateaus at less than 0.05 (service delivery outcome/indicator) with increased funding. Refugee hosting districts have on average higher service delivery outcomes than their non-refugee hosting counterparts. This may be attributed to the additional external financing provided by mainly development partners to start-up and maintain services in refugee hosting communities. Low numbers of sub-counties are characterized by high service delivery outcomes, which quickly reduce with increased administrative units(sub-counties) up to about 20, and thereafter the service delivery composite indicator slightly increases with the number of sub-counties. The number of town councils have a gently increasing effect on the service delivery outcomes.

## Comparison of the predictive accuracy of the three approaches

Comparison of the three approaches was achieved by learning from a consistent training dataset using the subset of variables returned by each approach, and then evaluated using the testing dataset. The results are reported in Table 9.

Table 9 shows that the predictive accuracy as presented by the mean square (MSE) or mean absolute error (MAE) for the four models is almost similar. This shows that, regardless of slightly different variables selected through regularization, any choice of the approach yields comparable predictive accuracy.

## Discussion

This research explores how the performance assessment of local governments can be enhanced through the construction of composite indicators, but also highlights potential statistical modelling approaches by associating the responsibility of the service outcomes to the local governments as centres of managing services. Of course, service delivery is multidimensional and so is its assessment meriting the construction of a composite indicator within a contextualized theoretical framework.

Generally, the composite score of service delivery is moderate, averaging slightly around 50% (on a range of 0–100%). The beta regression model identified four significant factors at 5% level of significance that may be clustered into three namely, i) financial (central government funding from the treasury and other funding from central government agencies), ii) administrative size (number of sub-counties with a negative relationship), and iii) geographical size (land area with a negative relationship). Due to group structures in the variables [56] and the nature of sparse models, the variables could be regressed as a group [57]. More sub-counties predict worse performance, suggesting a need to streamline administrative units or improve governance at sub-county levels; and geographic size: larger district local governments face service delivery challenges, potentially due to logistical constraints in supervision and monitoring. Inadequate oversight by central government entities in those regions may contribute to subpar service delivery. Of course, low financial inputs in interaction with other predictors exacerbate a cycle of low service delivery outcomes. With low financial resources, a higher geographical size may present infrastructural mobility challenges for effective implementation and monitoring of the operations of service delivery points. Indeed, budgetary investment enhances both the quantity and occasionally the quality of services that a provider can deliver; but, this is also contingent upon the government's capacity to generate, administer, and utilize public monies efficiently [58]. These findings regarding budgetary limitations correspond with positions that districts local governments in Uganda are *inter alia*, contending with insufficient local financial resources to attain satisfactory service delivery levels [59]. They assert that local governments are incapable of effectively mobilizing sufficient local financial resources, and their financing continues to depend significantly on central government grants which have been reduced overtime. In addition, the results of this study suggest that the ongoing fragmentation of local governments in Uganda exerts pressure on administrative capacity and financial systems, which detrimentally depletes resources that could otherwise be allocated for service delivery [7]. In our view, these significant predictors are subject to the discretion of local government policymakers; i) through the rationalization of cost centres/sub-counties, and ii) the enhancement and efficient administration of local government revenues. Based on the model results, it is suggested that increases in financial allocation to these three sectors should be made, after careful consideration and meta-analysis of the key

**Table 9. Comparison of predictive performance of the four approaches.**

| Approach | RMSE | MSE | MAE | AIC | BIC |
|---|---|---|---|---|---|
| Beta regression with varying phi | 0.0540 | 0.0029 | 0.0431 | −205.4307 | −171.7032 |
| Beta regression with fixed phi | 0.0529 | 0.0028 | 0.0427 | −203.1853 | −187.4458 |
| Random Forest | 0.0555 | 0.0031 | 0.0434 | – | – |
| Generalized additive model | 0.0517 | 0.0027 | 0.0421 | −217.6773 | −195.5015 |

sub-programs that have been proved to increase service delivery gains within these sectors. In line with the recommendation offered by Ggoobi and Lukwago (2019), streamlining of budget execution process for sectors with decentralized functions should be considered [6].

In this study we utilized the beta regression model with a varying dispersion parameter, as it effectively manages doubly bounded data such as indices. The inclusion of the precision parameter offers additional insights beyond the mean model, allowing for the assessment of predictors' impact on the outcome's variability. Moreover, prior research has demonstrated that the incorporation of the precision parameter results in the production of efficient coefficients, a valuable characteristic in the natural sciences. To address the necessity of balancing over-fitting and under-fitting, model regularization was implemented, resulting in the selection of significant variables. RF regression, utilizing machine learning techniques, is equally effective and identifies two variables as significant, like those indicated by beta regression model. This is the administrative size of the local government in form of the number of sub-counties and central government funding from the treasury to the district local governments. A generalized additive model enhances the analysis by bringing to the fore the non-linear predictor functions. We believe that the suggested statistical modelling approach, which employs beta regression with a varying precision parameter, followed by variable selection and bias correction, yields variables that elucidate service delivery outcomes via the composite indicator. Moreover, the application of a random forest in this domain in Uganda, when utilized alongside beta regression, may yield comparable variables that merit prioritization by policymakers. If the desire is to model non-linear covariate effects, GAM is a better approach as it reveals the trajectory of the covariates. Yes, the three statistical approaches yielded similar predictive accuracy (RMSE ≈0.05; MAE ≈ 0.042). Collectively, these instruments provide a solid empirical foundation for formulating evidence-based changes to improve public service delivery in decentralized environments. Methodologically, beta regression with varying dispersion stands out for its ability to capture mean effects and precision. Nonetheless, machine learning random forest and GAM approaches provide complementary strengths.

## Study limitations

The primary constraints of this study pertain to the insufficiency of elementary indicators and data points for evaluation. Due to the lack of accurate data at the sub-national level, fewer service sectors than required were incorporated into the model. The composite indicator would be more representative if it incorporated data from all tiers of the established theoretical framework, particularly regarding service processes. The dynamic nature of service delivery may introduce biases in the study, as it relied on annual data series, perhaps overlooking crucial service characteristics that fluctuate more frequently.

## Conclusion

This study enhances the current literature by addressing performance assessment gaps related to local government service delivery in developing countries, particularly the inadequacy of existing measures that only reflect disparate elementary indicators. This study illustrates that a composite indicator, which includes elementary indicators related to service inputs, processes, and outcomes while also considering quality, quantity, and equity, can be effectively integrated into a unified quantification framework to offer a comprehensive assessment of performance. It is demonstrated that no single statistical method is inherently superior; rather, methods can be effectively combined, with their accuracy and interpretability contingent upon the researcher's analytical objectives. Researchers examining local government performance should construct a composite indicator and select from one of the three tested statistical approaches (parametric, semi-parametric, or machine learning) to model differentials among statistical units. This study statistically shows the efficacy of beta regression with dispersion modeling for bounded response variables, such as composite indicator scores. Additionally, the significance of model regularization and diagnostics, particularly in robust predictive modeling is shown. The inclusion of a penalty term in the loss function facilitates regularization methods, which encourage the creation of simpler, more parsimonious models that demonstrate enhanced generalizability and improved performance in practical applications.

Future research planning should develop a composite indicator of service delivery that includes additional sectors such as justice, transport, and sanitation. This research would benefit from the compilation of additional elementary indicators derived from the same data generation approach to improve comparability. The incorporation of more frequent data, particularly from non-traditional sources such as social media, newspapers, and satellite imagery, would enhance the relevance and timeliness of the composite indicator as a performance assessment tool for district local governments. This approach would also empower citizens to hold policymakers accountable. The state of service delivery is dynamic and requires use of more frequent data sets. Future research should examine the application of grouped regression in the context of beta regression, particularly focusing on the use of cross-validation to evaluate model accuracy, given the group structures associated with the identified predictors of service delivery at the local government level.

## Supporting information

**S1 Table. Elementary indicators used.**
(DOCX)

**S2 Table. Cross frequencies among categorical variables.**
(DOCX)

**S3 Table. Correlation matrix for the continuous variables.**
(DOCX)

**S4 Table. Output of application of GAM.**
(DOCX)

**S1 Fig. OOB error rate for RF Vs number of trees grown.**
(DOCX)

**S2 Fig. Variable importance plot returned by RF.**
(DOCX)

**S3 Fig. GAM model diagnostic plots.**
(DOCX)

## Author contributions

**Conceptualization:** Hillary Muhanguzi.

**Formal analysis:** Hillary Muhanguzi.

**Methodology:** Hillary Muhanguzi.

**Supervision:** Francesca Bassi, Yeko Mwanga, James Wokadala.

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
