## [Decision Letter · Decision Letter 0]

26 Aug 2025

Dear Dr. Muhanguzi,

Thank you for submitting your manuscript to PLOS ONE. After careful consideration, we feel that it has merit but does not fully meet PLOS ONE’s publication criteria as it currently stands. Therefore, we invite you to submit a revised version of the manuscript that addresses the points raised during the review process.

We look forward to receiving your revised manuscript.

Kind regards,

José Antonio Ortega, Ph.D.

Academic Editor

PLOS ONE

Journal Requirements:

2. In the online submission form, you indicated that:

“The data are sourced from different publications and collated by the corresponding author in one file. It can be availed by a request on e-mail”

3. Uploaded as supplementary information.

3. We notice that your supplementary figures (A.3 to A.5) are included in the manuscript file. Please remove them and upload them with the file type 'Supporting Information'. Please ensure that each Supporting Information file has a legend listed in the manuscript after the references list.

4. We notice that your supplementary figures are uploaded with the file type 'Figure'. Please amend the file type to 'Supporting Information'. Please ensure that each Supporting Information file has a legend listed in the manuscript after the references list.

5.If the reviewer comments include a recommendation to cite specific previously published works, please review and evaluate these publications to determine whether they are relevant and should be cited. There is no requirement to cite these works unless the editor has indicated otherwise. 

**Additional Editor Comments:**

Two experts have reviewed your manuscript suggesting improvements on the purpose of the paper, missing methodological elements and other minor changes, concurring that the manuscript can comply with PLOS ONE requirements after these changes. Note that reviewer 1 suggests APA citations, but this is not PLOS ONE policy. The numbered system you are using is correct. You should just make sure that the references are in the appropriate format and complete.

Reviewers' comments:

Reviewer's Responses to Questions

**Comments to the Author**

1. Is the manuscript technically sound, and do the data support the conclusions?

Reviewer #1: Yes

Reviewer #2: Yes

2. Has the statistical analysis been performed appropriately and rigorously?

Reviewer #1: Yes

Reviewer #2: Yes

3. Have the authors made all data underlying the findings in their manuscript fully available?

Reviewer #1: Yes

Reviewer #2: Yes

4. Is the manuscript presented in an intelligible fashion and written in standard English?

Reviewer #1: Yes

Reviewer #2: Yes

Reviewer #1: Major feedback to authors:

1) The authors should provide more detail around why they chose to include three different models in the manuscript. Although substantial discussion is given to the statistical advantages and disadvantages of different models, the purpose of presenting all three is not explicitly stated. If the intent is to demonstrate robustness or to illustrate how different functional forms yield distinct policy insights, that rationale should be stated explicitly in the introduction. Conversely, if one model is preferred or if the models should be used in certain circumstances, the authors should justify why the other two are still included, despite producing broadly similar results.

2) The authors could more clearly explain the methods for creating the composite indicator. The explanation of how to normalize and adjust the indicators is clear but the process by which the three major domains are combined is unclear. To create the final composite indicator, the authors may take arithmetic mean or use a more complex method - I was unable to understand the exact final step in creating the composite indicator from their description.

3) The authors could further justify and explain the selection of the components of the service delivery composite indicator. While the identified components (health, water, and education) are important parts of service delivery, they are not a complete set of government responsibilities and exclude other areas of service delivery (roads, electricity, policing, and more). Choosing not to include these other aspects is a slight weakness and could potentially bias the model. This could be included as a limitation in the discussion section, acknowledging that the composite indicator aims to capture major aspects of service delivery but may be incomplete.

4) The authors could state whether they considered other indicators for inclusion in the health dimension. Most of the indicators included in the health domain are indicators of infrastructure or total quantity of services, and not the quality of services provided. There may not be sub-nationally representative data that measures quality and, if so, this should be stated.

5) The authors should acknowledge the limited explanatory power of their model. Although the model finds significant effects, the overall R-squared for the Beta model is 0.32, suggesting a substantial degree of variation remains unexplained by the covariates included. A sentence in the discussion section could address this limitation.

6) The authors explain that Local Government Act devolved responsibilities to lower levels but they could explain the exact degree and extent of the authority of LGs, such as what powers the LGs have to influence service delivery of the three core domains. This should ideally be added to the introduction.

Minor revisions:

Throughout the paper, the authors refer to other sources with a numeric. If included in the text, it should say the name of the author, e.g. “as shown by Jackson et al” rather than “as shown by (3)”. This may just be a formatting issue.

Line 12 should read: “The predictive accuracies of these approaches as established through the mean square error were found to be similar..”

Line 49, vague. “Recently, several services have been rendered by non-state actors as private businesses or on behalf of the State.” Please give examples or sources.

Line 52, word choice. Recommend changing to: “The Ugandan Government has been focused on improving”

Line 58, opinion. “Uganda has encountered both commendations and frustrations in service delivery in almost similar proportions.” Revise to more neutral phrasing, such as Uganda has encountered successes and challenges in service delivery (removing commentary on proportions).

Line 59, word choice. “These conflicting positions are supported by” should be revised to “These various issues are evidenced by”

Line 76, word choice. It’s not clear what is meant by “justified tagging”, please clarify.

Line 77-78, word choice. Revise “it remains unknown to the applicable statistical approaches for revealing the service delivery differentials” to “the appropriate statistical approach for revealing service delivery differentials remains unknown”.

Line 114, word choice: Revise to a less absolute statement, such as “this can be detrimental and may result in overfitting”

Line 116, vague. The authors state “Researchers such as (18) note that statistical analyses are essential yet highly complex in obtaining predictions, discoveries, and conclusions.” This sentence adds very little. Remove or make more clear what point is being made.

Line 129 should read: “the linear model is premised on other assumptions, such as that”

Line 283, word choice. The line says “As recommended by (40), descriptive for the variables were obtained as a search for pattern and structure of the data”. Please reword, the meaning is unclear.

Line 342, should read: “To perform the random forest algorithm”

Line 599, should read: “central government grants, which have been reduced over time.”

Reviewer #2: This study focuses on service delivery by local governments in Uganda. It constructs a composite indicator and applies multiple models (e.g., Beta regression, random forest) to analyze influencing factors. The research design aligns with the practical needs of public management, and the data processing and methodological application demonstrate basic rigor, providing a practical reference for the assessment of local governance in Uganda. However, there is room for improvement in terms of theoretical linkage, methodological details, result interpretation, and policy extension. Further refinement is needed to meet the requirements of PLOS ONE for research quality and contribution.

Specific Suggestions

Major Comments

1.The construction of the composite indicator refers to the OECD process but fails to link with classic theories of public service delivery. The selection of indicators and weighting schemes lacks in-depth integration with theoretical goals such as “fairness and efficiency” in service delivery.

Suggestion: Supplement the elaboration of the theoretical framework. Explain how the indicators correspond to the dimensions of “service inputs, processes, and outcomes.” Combine public service assessment theories to justify the rationality of the weighting schemes, providing a stronger theoretical basis for method selection.

2.The literature primarily focuses on domestic policies in Uganda, with insufficient references to similar international studies. It is difficult to highlight the theoretical contribution of this research in the field of “local government service governance.”

Suggestion: Supplement comparisons with international literature and analyze the uniqueness of the Ugandan case.

3.The analysis of the impact of key variables merely stays at the description of “significantly positive/negative correlation” and fails to explain the mechanisms in combination with the actual governance context of Uganda. For categorical variables, only frequency statistics are provided, without in-depth analysis of their impact paths on service delivery.

Suggestion: Explain the influence mechanisms of variables by combining details of the operation of local governments in Uganda.

4.The conclusion mentions “providing references for policymakers” but does not propose specific plans based on key findings.

Suggestion: Based on the model results, refine policy recommendations.

5.The discussion of limitations only generally mentions that “indicator dimensions can be expanded and variables can be supplemented,” failing to address core deficiencies. The planning for future research directions is relatively vague and lacks specificity.

Suggestion: Supplement the analysis of key limitations, clarify improvement paths, and demonstrate the sustainability of the research.

Minor Comments

1.For Figure 1, adding textual annotations for key steps to lower the comprehension barrier for readers.

2.Strengthen the correlation analysis between Figure 2 and Table 2. For example, supplement the discussion on “the impact of variables with high skewness/kurtosis on the distribution pattern of the composite indicator” and provide further explanations.

3.For the“*”marking “Leptokurtic variable” in Table 2, clearly define the judgment criteria for kurtosis abnormalities in the table notes or main text.

4.For the sub-figure titles in Figure 2, supplement explanations of the data period and analysis unit to enhance the self-explanatory nature of the figures.

5.There are grammatical issues in the sentences on Line 306 and Line 595; revisions are required.

6.Further optimize the format and arrangement of figures and tables (e.g., Table 8) for better presentation.

**Do you want your identity to be public for this peer review?** For information about this choice, including consent withdrawal, please see our Privacy Policy

Reviewer #1: **Yes: ** Ruben Conner

Reviewer #2: No

---

## [Author Response · Author response to Decision Letter 1]

11 Sep 2025

Reviewer #1:

1) The authors should provide more detail around why they chose to include three different models in the manuscript. Although substantial discussion is given to the statistical advantages and disadvantages of different models, the purpose of presenting all three is not explicitly stated. If the intent is to demonstrate robustness or to illustrate how different functional forms yield distinct policy insights, that rationale should be stated explicitly in the introduction. Conversely, if one model is preferred or if the models should be used in certain circumstances, the authors should justify why the other two are still included, despite producing broadly similar results.

Thank you. In the revised article with changes highlighted, from Line 181 to 202, we provide explanations behind the choice of the three models. The rationale of enhancing the robustness of the study and mitigating the risk of results being artifacts of a single method is included. This is in addition to assertions of authors as alluded in Lines 159-162.

2) The authors could more clearly explain the methods for creating the composite indicator. The explanation of how to normalize and adjust the indicators is clear but the process by which the three major domains are combined is unclear. To create the final composite indicator, the authors may take arithmetic mean or use a more complex method - I was unable to understand the exact final step in creating the composite indicator from their description.

Thank you. In the revised article with changes highlighted, from Lines 366 to 437 we have summarized the methods used to build the composite indicator. We have included the decision taken at theoretical framework, data normalization, weighting, aggregation and the how we have chosen the scores to use as the dependent variable. We have included a table of the possible combination of choices tested to further clarify the steps in Fig 1.

3) The authors could further justify and explain the selection of the components of the service delivery composite indicator. While the identified components (health, water, and education) are important parts of service delivery, they are not a complete set of government responsibilities and exclude other areas of service delivery (roads, electricity, policing, and more). Choosing not to include these other aspects is a slight weakness and could potentially bias the model. This could be included as a limitation in the discussion section, acknowledging that the composite indicator aims to capture major aspects of service delivery but may be incomplete.

Thank you. The choice of the selection of the sectors and elementary indicators was informed by the legal framework for local governments in Uganda, and the limited availability of data. This is explained in Lines 395 to 404 under the exploratory data analysis sub-section.

4) The authors could state whether they considered other indicators for inclusion in the health dimension. Most of the indicators included in the health domain are indicators of infrastructure or total quantity of services, and not the quality of services provided. There may not be sub-nationally representative data that measures quality and, if so, this should be stated.

Thank you. Like in the previous comment, the limited availability of data for elementary indicators at district local government level informed the selection of indicators. This has been acknowledged in Lines 400 to 404.

5) The authors should acknowledge the limited explanatory power of their model. Although the model finds significant effects, the overall R-squared for the Beta model is 0.32, suggesting a substantial degree of variation remains unexplained by the covariates included. A sentence in the discussion section could address this limitation.

Thank you. The R-square is one of the metrics for assessing the regression models, however it’s not without weaknesses. A sentence is provided in the Lines 613-625 that the performance of beta regression may be assessed using diagnostic plots, but also compares with the results of the previous researchers obtained.

6) The authors explain that Local Government Act devolved responsibilities to lower levels but they could explain the exact degree and extent of the authority of LGs, such as what powers the LGs have to influence service delivery of the three core domains. This should ideally be added to the introduction.

Thank you, this has been included in the introduction, while drawing from the legislation governing local government service delivery. It is included in the Lines 99 to 101.

7) Minor revisions: Throughout the paper, the authors refer to other sources with a numeric. If included in the text, it should say the name of the author, e.g. “as shown by Jackson et al” rather than “as shown by (3)”. This may just be a formatting issue.

This has not been addressed, as this is the recommended referencing style by PLOS.

8) Line 12 should read: “The predictive accuracies of these approaches as established through the mean square error were found to be similar..”

This has been addressed in Line 15.

9) Line 49, vague. “Recently, several services have been rendered by non-state actors as private businesses or on behalf of the State.” Please give examples or sources.

This has been addressed in Lines 69 to 72

10) Line 52, word choice. Recommend changing to: “The Ugandan Government has been focused on improving”

This has been addressed in Line 75

11) Line 58, opinion. “Uganda has encountered both commendations and frustrations in service delivery in almost similar proportions.” Revise to more neutral phrasing, such as Uganda has encountered successes and challenges in service delivery (removing commentary on proportions).

This has been addressed in Line 81

12) Line 59, word choice. “These conflicting positions are supported by” should be revised to “These various issues are evidenced by”

This has been addressed in Line 82

13) Line 76, word choice. It’s not clear what is meant by “justified tagging”, please clarify.

This has been addressed in Line 112

14) Line 77-78, word choice. Revise “it remains unknown to the applicable statistical approaches for revealing the service delivery differentials” to “the appropriate statistical approach for revealing service delivery differentials remains unknown”.

This has been addressed in Lines 112 to 116

15) Line 114, word choice: Revise to a less absolute statement, such as “this can be detrimental and may result in overfitting”

This has been addressed in Line 155

16) Line 116, vague. The authors state “Researchers such as (18) note that statistical analyses are essential yet highly complex in obtaining predictions, discoveries, and conclusions.” This sentence adds very little. Remove or make more clear what point is being made.

This sentence has been removed

17) Line 129 should read: “the linear model is premised on other assumptions, such as that”

This has been addressed in Line 170

18) Line 283, word choice. The line says “As recommended by (40), descriptive for the variables were obtained as a search for pattern and structure of the data”. Please reword, the meaning is unclear.

This has been addressed in Lines 347 to 350.

19) Line 342, should read: “To perform the random forest algorithm”

This has been addressed in Line 480.

20) Line 599, should read: “central government grants, which have been reduced over time.”

This has been addressed in Line 789.

Reviewer #2:

This study focuses on service delivery by local governments in Uganda. It constructs a composite indicator and applies multiple models (e.g., Beta regression, random forest) to analyze influencing factors. The research design aligns with the practical needs of public management, and the data processing and methodological application demonstrate basic rigor, providing a practical reference for the assessment of local governance in Uganda. However, there is room for improvement in terms of theoretical linkage, methodological details, result interpretation, and policy extension. Further refinement is needed to meet the requirements of PLOS ONE for research quality and contribution.

Specific Suggestions Major Comments

1) The construction of the composite indicator refers to the OECD process but fails to link with classic theories of public service delivery. The selection of indicators and weighting schemes lacks in-depth integration with theoretical goals such as “fairness and efficiency” in service delivery.

Suggestion: Supplement the elaboration of the theoretical framework. Explain how the indicators correspond to the dimensions of “service inputs, processes, and outcomes.” Combine public service assessment theories to justify the rationality of the weighting schemes, providing a stronger theoretical basis for method selection.

Thank you. The theoretical framework in Lines 366 to 386, links to two paradigms of service delivery assessments and guides the selection of elementary indicators utilised in the building of the composite indicator.

2) The literature primarily focuses on domestic policies in Uganda, with insufficient references to similar international studies. It is difficult to highlight the theoretical contribution of this research in the field of “local government service governance.”

Suggestion: Supplement comparisons with international literature and analyze the uniqueness of the Ugandan case.

Thank you. This has been addressed in Lines 104 to 116 by making comparisons with the accessed local government performance assessment methods and providing rationale for the novel composite indicator constructed and modeled in this study.

3) The analysis of the impact of key variables merely stays at the description of “significantly positive/negative correlation” and fails to explain the mechanisms in combination with the actual governance context of Uganda. For categorical variables, only frequency statistics are provided, without in-depth analysis of their impact paths on service delivery.

Suggestion: Explain the influence mechanisms of variables by combining details of the operation of local governments in Uganda.

Thank you. This has been addressed in Lines 551 to 559 where the likelihood influence of categorical variables on service delivery has been mentioned. Additionally, the discussion of results explains the influence of the significant variables on service delivery as seen in Lines 774 to 798

4) The conclusion mentions “providing references for policymakers” but does not propose specific plans based on key findings.

Suggestion: Based on the model results, refine policy recommendations.

Thank you, the study results provide recommendations for researchers/statisticians as well as local government policy makers. For statisticians, the recommendation to use any of the approaches is presented in Lines 834 to 851. For local government staff, the recommendations on revenue management, reduction of fragmentation of administrative areas are highlighted in Lines 789 to 798

5) The discussion of limitations only generally mentions that “indicator dimensions can be expanded and variables can be supplemented,” failing to address core deficiencies. The planning for future research directions is relatively vague and lacks specificity.

Suggestion: Supplement the analysis of key limitations, clarify improvement paths, and demonstrate the sustainability of the research.

Thank you. The sustainability of the research can be shown by the identified strengths and gaps that could be addressed through future research. This has been concretized and specified in Lines 852 to 862.

6) Minor Comments For Figure 1, adding textual annotations for key steps to lower the comprehension barrier for readers.

This has been addressed by including subsection on theoretical framework to aggregation, lines 366 to 430.

7) Strengthen the correlation analysis between Figure 2 and Table 2. For example, supplement the discussion on “the impact of variables with high skewness/kurtosis on the distribution pattern of the composite indicator” and provide further explanations.

This has been addressed in lines 532 to 537.

8) For the“*”marking “Leptokurtic variable” in Table 2, clearly define the judgment criteria for kurtosis abnormalities in the table notes or main text.

This has been addressed in lines 519 to 524. In addition, the table note indicates the judgement criteria.

9) For the sub-figure titles in Figure 2, supplement explanations of the data period and analysis unit to enhance the self-explanatory nature of the figures.

This has been addressed in the sub-title Fig 2 by including the data period and local governments as units of analysis.

10) There are grammatical issues in the sentences on Line 306 and Line 595; revisions are required.

This has been addressed.

11) Further optimize the format and arrangement of figures and tables (e.g., Table 8) for better presentation.

Table 8, owing to its large size, has been transferred to the appendix and renamed appendix Table A.4.

---

## [Decision Letter · Decision Letter 1]

1 Oct 2025

Dear Dr. Muhanguzi,

We look forward to receiving your revised manuscript.

Kind regards,

José Antonio Ortega, Ph.D.

Academic Editor

PLOS ONE

**Journal Requirements:**

Reviewers' comments:

Reviewer's Responses to Questions

**Comments to the Author**

Reviewer #1: All comments have been addressed

Reviewer #2: All comments have been addressed

2. Is the manuscript technically sound, and do the data support the conclusions?

Reviewer #1: Yes

Reviewer #2: Yes

3. Has the statistical analysis been performed appropriately and rigorously?

Reviewer #1: Yes

Reviewer #2: Yes

4. Have the authors made all data underlying the findings in their manuscript fully available?

Reviewer #1: Yes

Reviewer #2: Yes

5. Is the manuscript presented in an intelligible fashion and written in standard English?

Reviewer #1: Yes

Reviewer #2: Yes

**Reviewer #1: ** Appreciate the care and attention in responding to reviewer comments. My comments have been well addressed. Recommending for publication!

**Reviewer #2:**  1.The study mentions that the construction of the composite indicator refers to the OECD process and integrates the "World Bank's Public Administration Production Function" and the "Public Service Assessment Methodology". However, it fails to clearly specify how these theories specifically guide indicator design: for instance, the corresponding relationship between the "service input-process-outcome" dimensions and the "student-teacher ratio" (input) in the education sector, "medical institution coverage rate" (process) in the health sector, and "safe drinking water access rate" (outcome) in the water sector. Although indicators are listed in Table A.1, their connection to theoretical dimensions is not established. It is recommended to supplement text to directly link core indicators to the dimensions in the theoretical framework, thereby enhancing the clarity of theoretical support through such connections.

2.The analysis of the impact paths of categorical variables is insufficient. Regarding the variable "whether a district is a refugee-hosting area", the study only mentions that "refugee-hosting areas have slightly higher service delivery scores" but fails to decompose the underlying mechanism: whether it is the direct effect of supplementary funding from UNHCR or the driven service demand caused by the concentration of refugees. Further analysis is needed by incorporating the interaction effects between variables.

3.The discussion section can be deepened. For example, while the study mentions that "indicator dimensions can be expanded", it can supplement one specific potential small dimension. This revision does not question the completeness of the existing indicator system; it only makes the discussion on limitations more concrete and avoids excessive generality. It is recommended to separate the conclusion and discussion sections, with reference to the "Research Articles typically consist of the following headings" on the PLOS ONE official website.

4.In "by (47)" at Line 371, the author’s name is missing after "by". In English academic writing, when citing "a methodology proposed by a certain author", it is necessary to clearly state "by [Author Name] (47)". Simply writing "by (47)" will result in ambiguity regarding the citation object. 5.In the sentence "were incorporate using distance-to-reference point data normalization" at Line 382, the verb form is incorrect; "incorporate" should be changed to "incorporated" (passive voice requires the past participle form).

6.The word order of "on average" and "slightly" at Line 740 can be optimized to avoid semantic redundancy.

7.There are errors in the citation format. Please refer to the latest published papers and the requirements on the official website for revision.

8.There are issues with the reference format. Please refer to the latest published papers and the requirements on the official website for correction.

9.Most of the literatures cited in the study are from 2021 or earlier, and relevant studies on service delivery of Ugandan local governments from 2022 to 2024 have not been included.

10.The paragraph format still needs adjustment, particularly regarding issues with first-line indentation.

**Do you want your identity to be public for this peer review?** For information about this choice, including consent withdrawal, please see our Privacy Policy

Reviewer #1: **Yes: ** Ruben Conner

Reviewer #2: No

---

## [Author Response · Author response to Decision Letter 2]

13 Oct 2025

How Comments Have Been Addressed

1. Please add a choropleth map of your variable

Thank you. As per your policy and advise, the map has been removed due to the inability to obtain permission from the original copyright holder.

2. The methods section can be shortened. You do not need to describe in detail regression, gam models, random forests and beta regression

Thank you. Some formulas have been removed (lines 260-272, and 201-210) and details summarized.

3. Note that gam models are SEMIPARAMETRIC.

Thank you. This has been corrected throughout the article.

4. Separate discussion and conclusion, and include a separate section on limitations.

Thank you. This has been addressed. The suggested limitations on data availability and dynamic nature of service performance have been included.

5. The study mentions that the construction of the composite indicator refers to the OECD process and integrates the "World Bank's Public Administration Production Function" and the "Public Service Assessment Methodology". However, it fails to clearly specify how these theories specifically guide indicator design: for instance, the corresponding relationship between the "service input-process-outcome" dimensions and the "student-teacher ratio" (input) in the education sector, "medical institution coverage rate" (process) in the health sector, and "safe drinking water access rate" (outcome) in the water sector. Although indicators are listed in Table A.1, their connection to theoretical dimensions is not established. It is recommended to supplement text to directly link core indicators to the dimensions in the theoretical framework, thereby enhancing the clarity of theoretical support through such connections.

Thank you. Lines 345-392 have been revised to show how the theories informed the theoretical framework ultimately informing the selection of elementary indicators and subsequent analyses. Table A.1 has been expanded to include a detailed explanation of the elementary indicators, particularly their linkage with the results chain/theoretical framework.

6. The literature primarily focuses on domestic policies in Uganda, with insufficient references to similar international studies. It is difficult to highlight the theoretical contribution of this research in the field of “local government service governance.” Suggestion: Supplement comparisons with international literature and analyze the uniqueness of the Ugandan case.

Thank you. This was addressed in lines 82-97. It shows that the Ugandan case builds a composite indicator, but also performs service delivery differentials across local governments

7. The analysis of the impact of key variables merely stays at the description of “significantly positive/negative correlation” and fails to explain the mechanisms in combination with the actual governance context of Uganda. For categorical variables, only frequency statistics are provided, without in-depth analysis of their impact paths on service delivery. Suggestion: Explain the influence mechanisms of variables by combining details of the operation of local governments in Uganda.

Thank you. This has been addressed in Lines 630 to 649 for beta regression. In addition, the discussion section explains the influence mechanisms of variables by combining details of the operation of local governments in Uganda.

8. The conclusion mentions “providing references for policymakers” but does not

propose specific plans based on key findings. Suggestion: Based on the model results, refine policy recommendations.

Thank you. Specific policy recommendations have been refined such as in Lines, 782-787. Additionally, recommendations for Statisticians or researchers have 837- 846.

9. The discussion of limitations only generally mentions that “indicator dimensions can be expanded and variables can be supplemented,” failing to address core deficiencies. The planning for future research directions is relatively vague and lacks specificity. Suggestion: Supplement the analysis of key limitations, clarify improvement paths, and demonstrate the sustainability of the research.

Thank you. A subsection on study limits has covered the known limitations, which mainly rotate around inadequate data. The planning for future studies in light of data limitations has been addressed in lines 837 – 846, for example the use of non-traditional data sources, use of more frequent data, etc.

10. For Figure 1, adding textual annotations for key steps to lower the comprehension barrier for readers.

Thank you. Narrative of Fig 1 has been included in Lines 334 to 343

11. Strengthen the correlation analysis between Figure 2 and Table 2. For example, supplement the discussion on “the impact of variables with high skewness/kurtosis on the distribution pattern of the composite indicator” and provide further explanations.

Thank you. A column has been added onto Table 2 to show that whereas many conceivable scenarios for computation of the CI were tested, the scores for only one approach were utilised based on the stability test. It is these figures that were used in Figure 2 to plot the distributions and further analyses.

12. For the “*” marking “Leptokurtic variable” in Table 2, clearly define the judgment criteria for kurtosis abnormalities in the table notes or main text.

Thank you. The judgement criteria have been included as a Table note but also in the text in Lines 520-522.

13. For the sub-figure titles in Figure 2, supplement explanations of the data period and analysis unit to enhance the self-explanatory nature of the figures

Thank you, the figure caption includes the data period and unit of analysis

14. There are grammatical issues in the sentences on Line 306 and Line 595; revisions are required. There are issues with the reference format. Please refer to the latest published papers and the requirements on the official website for correction.

Thank you. All grammatical errors and referencing styles have been addressed in line with the PLOS One requirements.

15. The paragraph format still needs adjustment, particularly regarding issues with first-line indentation

Thank you. This has been addressed throughout the paper.

16. Most of the literatures cited in the study are from 2021 or earlier, and relevant studies on service delivery of Ugandan local governments from 2022 to 2024 have not been included.

Thank you. Some of the accessed recent literature have been included such as 42, 40, 15 and 14.

---

## [Decision Letter · Decision Letter 2]

20 Nov 2025

Statistical approaches for service delivery differentials as assessed through a composite indicator: application to Ugandan local governments

PONE-D-25-20628R2

Dear Dr. Muhanguzi,

We’re pleased to inform you that your manuscript has been judged scientifically suitable for publication and will be formally accepted for publication once it meets all outstanding technical requirements.

Kind regards,

José Antonio Ortega, Ph.D.

Academic Editor

PLOS ONE

Additional Editor Comments (optional):

Both reviewers are satisfied with the revision. In my case, I believe adding a choropleth map would have added insight to the article, and it is feasible. PLOS ONE only asks to ensure if copyrighted material is being used but there exists available non-copyrighted maps of Uganda. Still, it is the authors option, and this does not preclude the article from publication.

Reviewers' comments:

Reviewer's Responses to Questions

**Comments to the Author**

Reviewer #2: All comments have been addressed

2. Is the manuscript technically sound, and do the data support the conclusions?

Reviewer #2: Yes

3. Has the statistical analysis been performed appropriately and rigorously?

Reviewer #2: Yes

4. Have the authors made all data underlying the findings in their manuscript fully available?

Reviewer #2: Yes

5. Is the manuscript presented in an intelligible fashion and written in standard English?

Reviewer #2: Yes

Reviewer #2: Yes, the manuscript has made satisfactory revisions. The authors have addressed the key points raised previously, such as clarifying the indicator construction logic and adjusting format issues. The study's methodology and conclusions are now more coherent, and there are no obvious concerns regarding dual publication, research ethics, or publication ethics. It is recommended for acceptance.

**Do you want your identity to be public for this peer review?** For information about this choice, including consent withdrawal, please see our Privacy Policy

Reviewer #2: No

---

## [Editor Report · Acceptance letter]

PONE-D-25-20628R2

PLOS ONE

Dear Dr. Muhanguzi,

I'm pleased to inform you that your manuscript has been deemed suitable for publication in PLOS ONE. Congratulations! Your manuscript is now being handed over to our production team.

Kind regards,

on behalf of

Dr. José Antonio Ortega

Academic Editor

PLOS ONE